# Musashi proteins are post-transcriptional regulators of the epithelial-luminal cell state

Yarden Katz[1,2,4†], Feifei Li[3†], Nicole J Lambert[4], Ethan S Sokol[2,4], Wai-Leong Tam[2], Albert W Cheng[2,4], Edoardo M Airoldi[5,6], Christopher J Lengner[7,8], Piyush B Gupta[2,4], Zhengquan Yu[3*], Rudolf Jaenisch[2,4*], Christopher B Burge[4*]

[1]Department of Brain and Cognitive Sciences, Massachusetts Institute of Technology, Cambridge, United States; [2]Whitehead Institute for Biomedical Research, Cambridge, United States; [3]State Key Laboratories for Agrobiotechnology, College of Biological Sciences, China Agricultural University, Beijing, China; [4]Department of Biology, Massachusetts Institute of Technology, Cambridge, United States; [5]Department of Statistics, Harvard University, Cambridge, United States; [6]The Broad Institute, Cambridge, United States; [7]Department of Animal Biology, School of Veterinary Medicine, University of Pennsylvania, Philadelphia, United States; [8]Institute for Regenerative Medicine, University of Pennsylvania, Philadelphia, United States

**Abstract** The conserved Musashi (Msi) family of RNA binding proteins are expressed in stem/progenitor and cancer cells, but generally absent from differentiated cells, consistent with a role in cell state regulation. We found that Msi genes are rarely mutated but frequently overexpressed in human cancers and are associated with an epithelial-luminal cell state. Using ribosome profiling and RNA-seq analysis, we found that Msi proteins regulate translation of genes implicated in epithelial cell biology and epithelial-to-mesenchymal transition (EMT), and promote an epithelial splicing pattern. Overexpression of Msi proteins inhibited the translation of Jagged1, a factor required for EMT, and repressed EMT in cell culture and in mammary gland *in vivo*. Knockdown of Msis in epithelial cancer cells promoted loss of epithelial identity. Our results show that mammalian Msi proteins contribute to an epithelial gene expression program in neural and mammary cell types.

*For correspondence: zyu@cau.edu.cn (ZY); jaenisch@wi.mit.edu (RJ); cburge@mit.edu (CBB)

†These authors contributed equally to this work

## Introduction

During both normal development and cancer progression, cells undergo state transitions marked by distinct gene expression profiles and changes in morphology, motility, and other properties. The Epithelial-to-Mesenchymal Transition (EMT) is one such transition, which is essential in development and is thought to be co-opted by tumor cells undergoing metastasis (*Polyak and Weinberg, 2009*). Much work on cell state transitions in both the stem cell and cancer biology fields has focused on the roles that transcription factors play in driving these transitions (*Polyak and Weinberg, 2009*; *Lee and Young, 2013*), such as the induction of EMT by ectopic expression of the transcription factors Snail, Slug, or Twist (*Mani et al., 2008*).

Recent work has shown that RNA-binding proteins (RBPs) also play important roles in cell state transitions, by driving post-transcriptional gene expression programs specific to a particular cell state. The epithelial specific regulatory protein (ESRP) family of RBPs are RNA splicing factors with epithelial tissue-specific expression whose ectopic expression can partially reverse EMT (*Warzecha et al., 2009*; *Shapiro et al., 2011*). RBPs have also been implicated in other cell state transitions, such as

**eLife digest** All living things start life as a single cell, but many organisms develop into a collection of different, specialized cells. Most of the cells in an organism can only divide to make more of the same type of cell; however, stem cells are different because they can 'differentiate' and develop into several different cell types.

A key step in the development of an embryo is called the epithelial-to-mesenchymal transition, in which an epithelial cell—a cell type that normally lines body surfaces and cavities—begins to crawl away from the tissue it is in and starts to differentiate. This transition also allows cancer cells to leave tumors and spread around the body, in a process known as metastasis.

In mammals, two proteins called Musashi1 and Musashi2 are abundant in stem cells and brain cancers, but are rarely found in specialized tissues and cells. Katz, Li et al. now find that the Musashi proteins are also often overexpressed in human breast, lung, and prostate tumors. In addition, Musashi proteins are much less abundant in cells that have completed an epithelial-to-mesenchymal transition.

When Katz, Li et al. artificially reduced the amounts of Musashi proteins in breast cancer cells, the cells migrated and dispersed, as if becoming mesenchymal cells. Furthermore, many of the genes normally used in epithelial cells were switched off. In comparison, artificially increasing the levels of Musashi proteins halted the movement of mesenchymal cells and led to increased levels of genes used in epithelial cells, as if they were reverting to epithelial cells. Therefore, it appears that the Musashi proteins prevent epithelial cells from developing mesenchymal properties.

Katz, Li et al. investigated how Musashi proteins work at the molecular level by studying neural and mammary cells in mice. This revealed that Musashi proteins control the steps that lead to the epithelial-to-mesenchymal transition by binding to the tail end of the RNA molecules that include the instructions to make certain proteins. This affects how often these proteins can be made from the RNA molecules. Katz, Li et al. suggest that Musashi proteins may similarly control the behavior of progenitor and stem cells in many other tissues as well; however, further study is needed to confirm this.

reprogramming of somatic cells to induced pluripotent stem cells (iPSCs), which have the essential characteristics of embryonic stem cells (ESCs). For example, overexpression of the translational regulator and microRNA processing factor Lin28 along with three transcription factors is sufficient to reprogram somatic cells (*Yu et al., 2007*). The Muscleblind-like (Mbnl) family of RBPs promote differentiation by repressing an ESC-specific alternative splicing program, and inhibition of Mbnls promotes cellular reprogramming (*Han et al., 2013*). For ESRP, Lin28, and Mbnl proteins, the developmental or cell-type-specific expression pattern of the protein provided clues to their functions in the maintenance of epithelial, stem cell, or differentiated cell state.

The Musashi (Msi) family comprises some of the most highly conserved and tissue-specific RBPs, with *Drosophila Msi* expressed exclusively in the nervous system (*Nakamura et al., 1994*; *Busch and Hertel, 2011*). In mammals, the two family members *Msi1* and *Msi2* are highly expressed in stem cell compartments but are mostly absent from differentiated tissues. *Msi1* is a marker of neural stem cells (NSCs) (*Sakakibara et al., 1996*) and is also expressed in stem cells in the gut (*Kayahara et al., 2003*) and epithelial cells in the mammary gland (*Colitti and Farinacci, 2009*), while *Msi2* is expressed in hematopoietic stem cells (HSCs) (*Kharas et al., 2010*). This expression pattern led to the proposal that Msi proteins generally mark the epithelial stem cell state across distinct tissues (*Okano et al., 2005*), with HSCs being an exception. *Msi1* is not expressed in the normal adult brain outside a minority of adult NSCs but is induced in glioblastoma (*Muto et al., 2012*).

Msi proteins affect cell proliferation in several cancer types. In glioma and medulloblastoma cell lines, knockdown of *Msi1* reduced the colony-forming capacity of these cells and reduced their tumorigenic growth in a xenograft assay in mice (*Muto et al., 2012*). Msi expression correlates with HER2 expression in breast cancer cell lines, and knockdown of Msi proteins resulted in decreased proliferation (*Wang et al., 2010*). These observations, together with the cell-type specific expression of Msi proteins in normal development, suggested that Msi proteins might function as regulators of cell state, with potential relevance to cancer.

Msi proteins have been proposed to act as translational repressors of mRNAs—and sometimes as activators (*MacNicol et al., 2011*)—when bound to mRNA 3' UTRs, and were speculated to affect

pre-mRNA processing in *Drosophila* (*Nakamura et al., 1994*; *Okano et al., 2002*). However, no conclusive genome-wide evidence for either role has been reported for the mammalian Msi family. Here, we aimed to investigate the roles of these proteins in human cancers and to gain a better understanding of their genome-wide effects on the transcriptome using mouse models.

## Results

### Msi genes are frequently overexpressed in multiple human cancers

To obtain a broad view of the role Msis might play in human cancer, we surveyed the expression and mutation profiles of Msi genes in primary tumors using genomic and RNA sequencing (RNA-Seq) data from The Cancer Genome Atlas (TCGA) (*Cancer Genome Atlas Network., 2012*). To determine whether Msi genes are generally upregulated in human cancers, we analyzed RNA-Seq data from five cancer types for which matched tumor-control pairs were available. In these matched designs, a pair of RNA samples was obtained in parallel from a single patient's tumor and healthy tissue-matched biopsy, thus minimizing the contribution of individual genetic variation to expression differences. We observed that *Msi1* was upregulated in at least 40% of breast, lung, and prostate tumors, while *Msi2* was upregulated in at least 50% of breast and prostate tumors (*Figure 1A*, top). Overall, *Msi1* or *Msi2* were significantly upregulated in matched tumor-control pairs for 3 of the 5 cancer types, compared to control pairs. Kidney tumors showed the opposite expression pattern, with *Msi1* and *Msi2* downregulated in a majority of tumors and rarely upregulated, and in thyroid cancer neither *Msi1* nor *Msi2* showed a strong bias towards up- or down-regulation (*Figure 1A*, top). In breast tumors, a bimodal distribution of *Msi1* expression was observed, with a roughly even split between up- and down-regulation of *Msi1*, consistent with the idea that *Msi1* upregulation might be specific to a subtype of breast tumors. The bimodality of *Msi1* expression was not seen when comparing control pairs, so is not explained by general variability in *Msi1* levels (*Figure 1A*, bottom, solid vs dotted lines).

Examining genome sequencing data from matched tumor-control pairs across nine diverse cancer types, we found that *Msi1* and *Msi2* were not significantly mutated in most of these cancers (*Figure 1B*). One notable exception was kidney cancer (KIRC), where non-silent mutations in *Msi1* were significantly overrepresented, detectable in 9% of tumors (ranked in the 99[th] percentile of mutations per gene in this cancer) (*Figure 1—figure supplement 1A*). This observation, together with the lower Msi mRNA levels observed in matched kidney tumors (*Figure 1A*), is consistent with a model in which loss of Msi function is selected for in kidney tumor cells, either as a result of downregulation or mutation. The observation that *Msi1/Msi2* was not significantly mutated in most tumors but are overexpressed in several tumor types (including glioblastoma) makes their profile more similar to oncogenes like FOS or HER2, than to tumor suppressors like PTEN and TP53, which tend to have the opposite pattern (*Verhaak et al., 2010*; *Cancer Genome Atlas Network., 2012*) (*Figure 1B*).

### Msi expression marks an epithelial-luminal state and is downregulated upon EMT

To determine whether Msi overexpression is specific to a particular cancer cell state, we focused on breast cancer, where tumors with distinct properties can be robustly classified by gene expression (*Parker et al., 2009*; *Cancer Genome Atlas Network., 2012*). Unsupervised hierarchical clustering of matched tumor and control samples produced a nearly perfect separation of tumors from control samples, rather than clustering by patient/genome of origin (*Figure 1—figure supplement 1B*). We overlaid on top of our clustering a classification of samples into Normal, HER2+, Luminal A, Luminal B, and Basal states using RNA-Seq data to measure expression of the PAM50 gene set (*Parker et al., 2009*). Our clustering using all genes corresponded well to the PAM50 classification (*Cancer Genome Atlas Network., 2012*), separating most Luminal A from Luminal B tumors and showing a general grouping of HER2+ tumors (*Figure 1—figure supplement 1B*). Using this classification, we found that *Msi2* was highly expressed in Luminal tumors (*Figure 2A*). *Msi1* was more variable across tumor subtypes, often showing a bimodal profile, split between up- and down-regulation (*Figure 1A* and *Figure 2—figure supplement 1B*). *Msi2* expression was highest in Luminal B tumors, which are known to be more aggressive and highly proliferating (Ki67-high) than Luminal A types and are thought to share properties with epithelial mammary progenitor cells (*Das et al., 2013*). These observations prompted the hypothesis that Msi proteins might be localized to epithelial cells in breast cancer tumors. The splicing factors *Rbfox2* and *Mbnl1* were previously identified as regulators of EMT and are

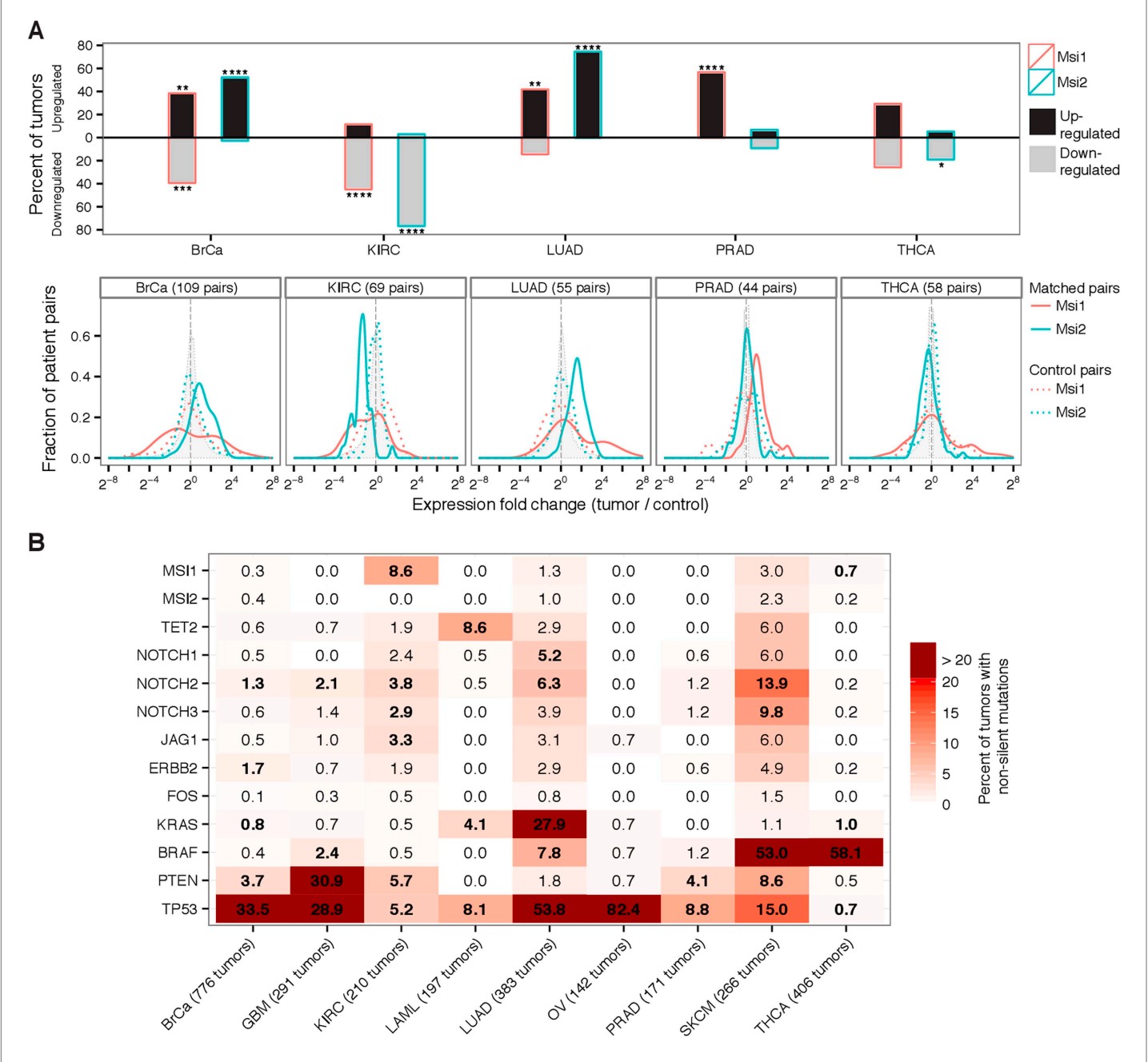

**Figure 1**. Msi genes are frequently overexpressed in breast, lung, and prostate cancer but downregulated in kidney cancer. (**A**) Top: percentage of matched tumor–control pairs with upregulated (black-fill bars) or downregulated (grey-fill bars) *Msi1* or *Msi2* in five cancer types with matched RNA-Seq data. Upregulated/downregulated defined as at least two-fold change in expression in tumor relative to matched control. Asterisks indicate one-tailed statistical significance levels relative to control pairs. Bottom: distribution of fold changes for *Msi1* and *Msi2* in matched tumor–control pairs (solid red and green lines, respectively) and in an equal number of control pairs (dotted red and green lines, respectively.) Shaded gray density shows the fold change across all genes. (**B**) Percentage of tumors with non-silent mutations in *Msi1/Msi2* and a select set of oncogenes and tumor suppressors across nine cancer types. Bold entries indicate genes whose mutation rate is at least two-fold above the cancer type average mutation rate.

The following figure supplement is available for figure 1:

**Figure supplement 1**. Analysis of *Msi1/Msi2* mutation and expression profiles in TCGA datasets.

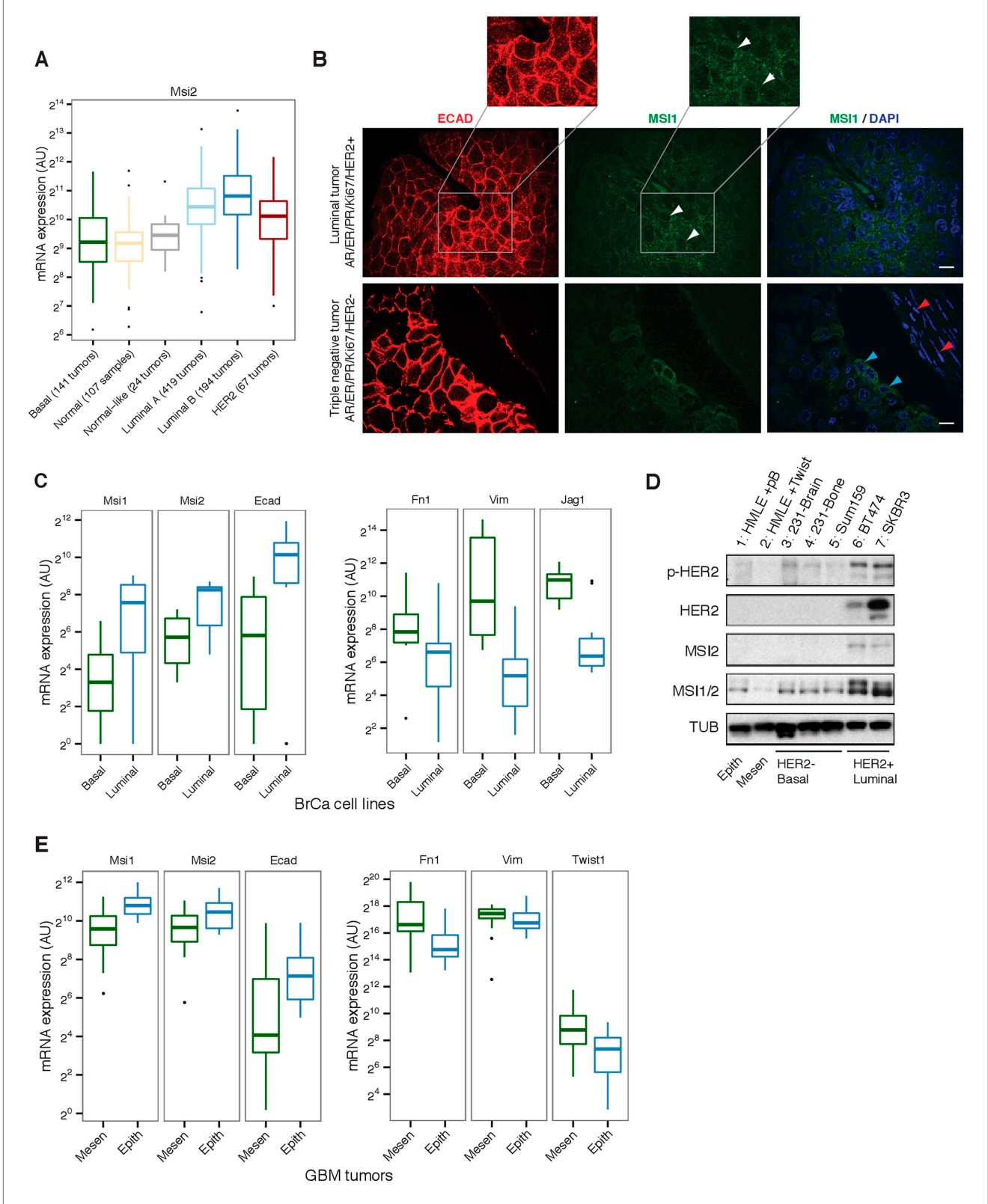

**Figure 2**. Msi is associated with the epithelial-luminal state in breast cancer. (**A**) mRNA expression of *Msi2* across different breast tumor types in TCGA RNA-Seq. (**B**) Immunofluorescence staining for Ecadherin (ECAD, red) and *Msi1* (MSI1, green). Top: luminal human breast tumor with high number of ECAD-positive cells. MSI1 shows primarily cytoplasmic localization (white arrowheads). Inset shows magnified version of ECAD and MSI staining.
*Figure 2. Continued on next page*

*Figure 2. Continued*

Bottom: triple negative, basal-like tumor. ECAD-positive cells showed strong cytoplasmic MSI1 stain (blue arrowheads) while ECAD-negative cells were MSI1-negative (red). Single confocal stacks shown, 10 μm scale. (**C**) mRNA expression of *Msi1*, *Msi2*, *Ecad*, *Fn1*, *Vim*, and *Jag1* in breast cancer cell lines by RNA-Seq (datasets are listed in *Supplementary file 1*). (**D**) Western blot for MSI1/2 (MSI1/2 cross react. antibody), MSI2, phosphorylated HER2 (p-HER2) and HER2 in panel of breast cell lines. 'HMLE + pB' indicates HMLE cells infected with pB empty vector, 'HMLE + Twist' indicates HMLE cells infected with Twist transcription factor to induce EMT. MDAMB231-derived metastatic lines (231-Brain, 231-Bone) and Sum159 are basal, HER2-negative cancer cell lines. BT474 and SKBR3 are HER2-positive, epithelial-luminal cancer cell lines. Epithelial-luminal (HER2-positive) lines show increased expression of Msi proteins compared with basal lines, and Twist-induced EMT reduces Msi expression. (**E**) mRNA expression of *Msi1*, *Msi2*, *Ecad*, *Fn1*, *Vim*, and *Twist1* in GBM tumors classified as mesenchymal (*n* = 20) or epithelial (*n* = 20) using an EMT gene signature.

The following figure supplements are available for figure 2:

**Figure supplement 1**. Expression of *Msi1/Msi2* in subtypes of breast cancer cell lines and breast cancer tumors.

**Figure supplement 2**. Expression of Rbfox2 (Rbm9) and Mbnl1 in subtypes of breast cancer tumors from TCGA.

upregulated during this transition (*Venables et al., 2013*). Using TCGA expression analysis, we confirmed that *Rbfox2* and *Mbnl1* are more highly expressed in luminal tumors compared with mesenchymal tumors, as predicted by their role in EMT (*Figure 2—figure supplement 2*).

To examine the expression and distribution of Msi proteins in tumors, we stained a panel of human breast cancer tumors for MSI1 and the epithelial marker E-cadherin (ECAD). MSI1 expression was predominantly cytoplasmic (*Figure 2B*, top panel). Across luminal tumors, MSI1 was co-expressed with ECAD (as in *Figure 2B*, top panel). In triple negative/basal-like tumors, a minority of ECAD-positive cells showed strong MSI1 staining, whereas ECAD-negative cells showed little to no expression (*Figure 2B*, blue and red arrowheads, respectively), supporting an association between Msi and epithelial cell state in tumors. Given the heterogeneity of human tumor samples, it is possible that the increased expression of Msi genes in luminal tumors (compared with basal) reflects the generally higher fraction of epithelial cells in these tumors.

To explore whether Msi expression is associated with a luminal as opposed to basal state in a more homogenous system, we analyzed RNA-Seq data for luminal and basal breast cancer cell lines generated by multiple independent labs (RNA-Seq data sets used are listed in *Supplementary file 1*). Gene expression profiles from the same cell lines generated independently tended to cluster together in unsupervised clustering (supporting consistency of data across labs), and overall the basal cell lines were distinguishable from the luminal lines (*Figure 2—figure supplement 1A*). Matching the pattern observed in primary tumors, we observed higher *Msi1* and *Msi2* expression in luminal breast cancer lines than in basal lines (*Figure 2C*, left panel). Expression of Fibronectin (*Fn1*), Vimentin (*Vim*), and Jagged1 (*Jag1*), which are associated with the basal/mesenchymal state (*Yamamoto et al., 2013*), had the opposite pattern, showing strong enrichment in basal over luminal lines (*Figure 2C*, right panel). The enrichments of these four genes for either the luminal or basal state were unusual when compared to the background distribution of these enrichments across all expressed genes (*Figure 2—figure supplement 1C*), indicating that these genes are strong indicators of the two states.

To further investigate the connection between Msi expression and EMT in breast cancer, we examined Msi expression in a panel of breast cancer-derived cell lines. Consistent with the RNA-Seq data from primary tumors, HER2+ epithelial cell lines expressed higher levels of *Msi1* and *Msi2* compared with HER2– lines (*Figure 2D*, lane 6 and 7). A standard cell culture model of EMT is the immortalized inducible-Twist human mammary epithelial (HMLE-Twist) cell line, which undergoes EMT when induced to express the transcription factor Twist (*Mani et al., 2008*). We found that *Msi1* was strongly downregulated in HMLE cells following Twist-induced EMT (*Figure 2D*), consistent with the epithelial-associated expression pattern of Msis in primary tumors (*Figure 2A–C*). Similarly, Msi protein expression was higher in luminal, HER2+ breast cancer lines (BT474, SKBR3 in *Figure 2D*) compared with basal HER2– breast cancer lines (brain and bone metastatic derivatives of MDAMB231, 231-Brain and 231-Bone, and SUM159 in *Figure 2D*).

We next asked whether the epithelial expression signature of Msis is present in other primary tumors. Given the established role of Msi proteins as regulators of Glioblastoma (GBM) cell growth and as markers of primary tumors (*Muto et al., 2012*), we examined whether there is a similar subtype expression pattern in GBM tumors from TCGA (*Verhaak et al., 2010*). We used an EMT gene

signature to rank GBM tumors from more epithelial to more mesenchymal, based on the similarity of each tumor's gene expression profile to that of cells undergoing EMT in culture (*Feng et al., 2014*). Using this ranking, we found that the top 20 most epithelial tumors expressed higher levels of Msi and epithelial markers like ECAD (*Figure 2E*). By contrast, the top 20 most mesenchymal tumors expressed lower levels of Msi and higher levels of mesenchymal markers like Fibronectin and Vimentin (*Figure 2E*). Thus, Msi expression is enriched in epithelial tumors in GBM as well, consistent with the results obtained in breast cancer tumors and cell lines.

Taken together, these results show that Msi genes are rarely mutated but frequently overexpressed across human cancers and are strong markers of the epithelial-luminal state. This pattern suggests that Msi proteins may play a role in the maintenance of an epithelial state and/or repression of EMT, in both breast and neural cell types. To better understand the molecular functions of Msi proteins, we turned to a controlled cell culture system.

## Genetic system for inducible overexpression and depletion of *Msi1/2* in NSCs

The upregulation of Msi genes in glioblastoma motivated the choice of NSCs as a system to study the molecular roles of Msi proteins, a cell type where both proteins are highly expressed in normal development, and where their target mRNAs are likely to be present. NSCs provide a well-characterized system for homogeneous cell culture (*Kim et al., 2003*), which is not always available for progenitor/stem cell types cultured from other primary tissues like the mammary gland, making NSCs grown in culture amenable to analysis by genome-wide techniques. Furthermore, the conserved expression of Msi genes in the nervous system and their reactivation in human glioblastoma suggests that molecular insights obtained in this system could be informative about the roles of Msi proteins in glioblastoma cells.

We cultured cortical NSCs from E12.5 embryos obtained from transgenic mice with a Dox-inducible *Msi1* or *Msi2* allele, and from double conditional knockout mice for *Msi1/Msi2*, whose deletion was driven by a Tamoxifen-inducible Cre (*Figure 3A*). These systems enabled robust overexpression or depletion of Msi proteins (*Figure 3B*) within 48–72 hr of induction. To study the effects of Msi depletion and induction on mRNA processing, expression, and translation, we used ribosome footprint profiling (Ribo-Seq) (*Ingolia et al., 2009*) and high-throughput sequencing of polyA-selected RNA (RNA-Seq) (*Mortazavi et al., 2008*) (*Figure 3A*).

## Overexpression of *Msi1* alters translation of targets without causing large changes in mRNA levels

When *Msi1* or *Msi2* were overexpressed, few significant changes in mRNA expression were observed after 48 hr (*Figure 3C*). This observation suggests that these factors do not directly impact transcription or mRNA stability/decay but leaves open possible effects on other steps in gene expression such as mRNA translation. To determine the genome-wide effects of Msi proteins on translation, we performed Ribo-Seq on *Msi1*-overexpressing cells and double knockout cells. Reads from these Ribo-Seq libraries showed the expected enrichment in coding exons relative to UTRs and introns, and yielded high scores in various quality control (QC) metrics (*Figure 3—figure supplement 1*). These QC metrics were highly consistent across libraries, supporting comparative analysis of the resulting data (*Figure 3—figure supplement 1*). To examine changes in translation, we computed 'Translational Efficiency' (TE) values for all protein-coding genes, a measure of ribosome occupancy along messages that is defined as the ratio of the ribosome footprint read density in the ORF to the RNA-seq read density. Examination of TEs across overexpression and knockout samples yielded a handful of genes with very large changes in ribosome occupancy (*Figure 3D*, 'Materials and methods').

## *Msi1* represses translation of Notch ligand Jagged1 and regulates translation of RBPs

Several genes exhibited substantial changes in their translation efficiency in response to overexpression of *Msi1*, including six genes with increased TE and three with reduced TE (*Figure 3D*). Genes with increased translation included the RNA processing factor *Prpf3/Prp3p*, a U4/U6 snRNP-associated factor, and genes involved in epithelial cell biology such as Kirrel3/NEPH2. Genes with repressed translation included: *Rbm22/Cwc2*, another splicing factor associated with U6 snRNP; *Dhx37*, an RNA helicase with possible role in alternative splicing (*Hirata et al., 2013*); and *Jag1*, a ligand of Notch receptors and an important regulator of Notch signaling. No change was detected in translation of previously

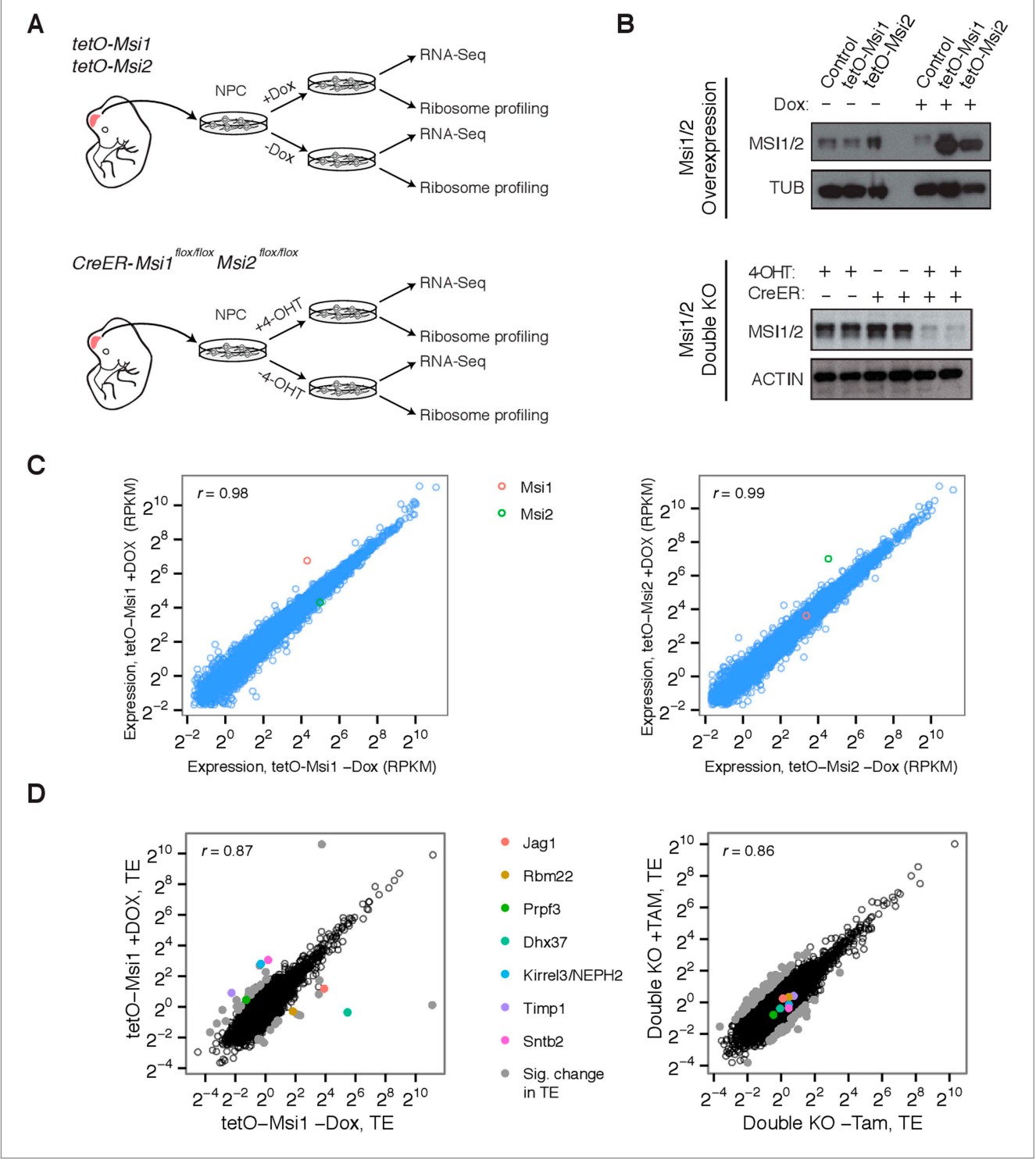

**Figure 3**. Genetic system for studying effects of Msi loss/gain of function on gene expression. (**A**) Experimental setup and use of *Msi1/2* inducible overexpression and conditional double knockout mice for derivation of neural stem cells, which were then used for ribosome profiling (Ribo-Seq) and mRNA sequencing (RNA-Seq). (**B**) Western blot analysis of Musashi overexpression and knockout in neural stem cells. Overexpression and conditional knockout cells were exposed to Dox and 4-OHT for 72 hr, respectively. (**C**) mRNA-Seq expression values (RPKM) scatters between *Msi1* overexpressing cells and controls (left), *Msi2* overexpressing cells and controls right (72 hr Dox). *Msi1/2* each robustly overexpressed with similar magnitude following Dox. (**D**) Comparison of translational efficiency (TE) values using Ribo-Seq on Msi1 overexpressing cells on Dox (72 hr) vs controls (left) and conditional knockout cells following 4-OHT for 48 hr (right). Colored points indicate select genes with large changes in TE.

*Figure 3. Continued on next page*

*Figure 3. Continued*

The following figure supplement is available for figure 3:

**Figure supplement 1**. Quality control metrics for Ribo-Seq libraries.

reported Msi target *Numb* (**Okano et al., 2002**), though Numb had low coverage of Ribo-Seq reads in NSCs, reducing our statistical power to detect regulation ('Materials and methods'). To explore whether the observed changes are mediated by direct protein binding to RNA targets, we mapped the RNA binding specificity of Msis.

## MSI1 shows high affinity for specific RNA motifs containing one or more UAGs

To determine sequence-specific RNA binding preferences of Msi proteins, we used 'RNA Bind-n-Seq' (RBNS) to obtain quantitative and unbiased measurement of the spectrum of RNA motifs bound by recombinant MSI1 protein in vitro (**Lambert et al., 2014**) (**Figure 4A**). For each 6mer, the 'R value' was defined as the occurrence frequency in libraries derived from MSI1-bound RNAs divided by the corresponding frequency in the input RNA library, and 6mer 'enrichment' was defined as the maximum R value observed across all protein concentrations. The fold enrichment profiles obtained by RBNS for the top five most enriched 6mers and five randomly chosen 6mers are shown in **Figure 4B**. Enriched 6mers exhibited similar enrichment profiles across concentrations, peaking in fold enrichment at concentrations typically between 16–64 nM (**Figure 4B**). To summarize the binding preferences of MSI1 from RBNS, we aligned the most enriched 6mers to generate a motif, which emphasizes that MSI1 binds predominantly to UAG-containing sequences, preferentially flanked by Us (**Figure 4C**). The MSI1 binding site (G/A)UAGU from a previous SELEX study was ~threefold enriched by RBNS, along with highly similar sequences, confirming binding under our assay conditions (**Imai et al., 2001**; **Ray et al., 2013**). Closer examination of the RBNS data revealed evidence for longer, higher-affinity motifs containing multiple UAGs with short intervening spacers (not shown).

Previous studies suggested that MSI1 binds 3′ UTR regions of mRNAs to regulate translation (**Okano et al., 2005**). We calculated the density of RBNS-enriched 6mers in 3′ UTR regions genome-wide and ranked genes by the density of enriched 6mers in their 3′ UTR ('Materials and methods'). We observed that the 3′ UTR of *Jag1*—which is translationally repressed by Msi (**Figure 3D**)—contains a moderately high density of RBNS-enriched 6mers, ranking in the 85th percentile of all 3′ UTRs (**Figure 4D**). To ask whether Msi proteins can directly bind the *Jag1* mRNA and test the RBNS motif, we selected two regions of the *Jag1* 3′ UTR that contained the highest density of RBNS-enriched 6mers for in vitro analysis (**Figure 4B**, top). A gel-shift assay detected strong binding of RNAs representing both regions by recombinant Msi protein, with estimated $K_d$ values of 15 nM and 9 nM for regions 1 and 2, respectively (representative gel shifts are shown in **Figure 4—figure supplement 1**). Since both sequences contain UAGs (**Figure 4—figure supplement 1**), we hypothesized that the UAGs nucleate binding. Mutation of the UAG sites to UCC reduced binding to MSI1 protein by an order of magnitude or more in each case (**Figure 4E**), supporting a model where MSI1 binding occurs primarily at these sites.

Following Msi overexpression, the Ribo-Seq density of the *Jag1* coding region was reduced by ~fivefold, while its mRNA level was little changed, suggesting a predominant effect at the translational level (**Figure 4—figure supplement 2**). In double knockout cells, *Jag1* mRNA increased ~1.5-fold by RNA-Seq (**Figure 4—figure supplement 2**), with a similar increase in Ribo-Seq density, suggesting effects on message stability either in the absence of or as a consequence of translational derepression. Western blot analysis confirmed repression of JAG1 protein by *Msi1* overexpression (**Figure 4F**) and derepression in double knockout cells (**Figure 4G**). The high similarity between MSI1 and MSI2 proteins (over 70% identity at the amino acid level, with highly similar RNA recognition motifs) suggests similarity in function, and we confirmed that *Msi2* overexpression also repressed JAG1 protein expression by Western analysis (**Figure 4H**). To directly test the hypothesis that Msi proteins regulate *Jag1* translation via UTR binding, we constructed luciferase reporters for the *Jag1* 3' UTR and transfected these into 293T cells. Knockdown of *MSI1* or knockdown of both *MSI1* and *MSI2* increased luciferase expression in these cells, relative to mock knockdown treatments (**Figure 4—figure supplement 3**). This observation also indicates that Msi-dependent regulation of *Jag1* translation is conserved from murine

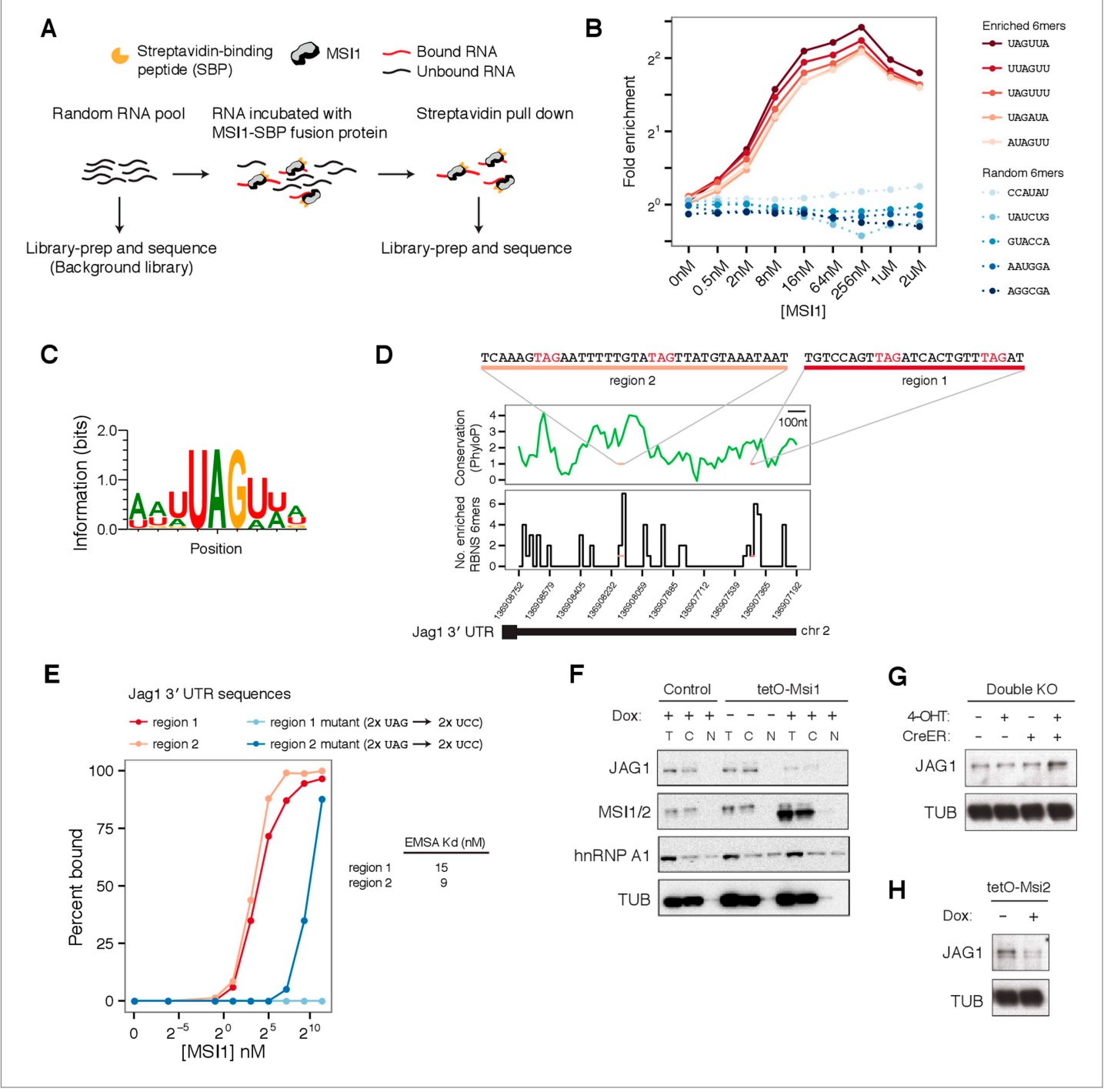

**Figure 4**. Profiling MSI1 binding preferences by RNA Bind-n-Seq. (**A**) Schemaic of Bind-n-Seq experiment for MSI1 protein. Increased concentrations of MSI1-SBP fusion protein incubated with random RNA pool, pulled by straptavidin pull-down, reverse-transcribed and sequenced. (**B**) Fold enrichment of top five enriched 6mers (red curves) and five randomly chosen 6mers (blue curves) across protein concentrations. (**C**) Binding motif for MSI1. Position-weight matrix generated by global alignment of top 20 enriched 6mers. (**D**) Two sites in *Jag1* 3' UTR, region 1 and region 2, containing a high density of enriched 6mers. Top: PhyloP conservation score for 3' UTR in 20 nt windows (based on UCSC vertebrates multiple alignment). Bottom: number of enriched 6mers from BNS in 20 nt windows of 3' UTR. (**E**) Percent binding of MSI1 protein to region 1 and region 2 (red curves) and mutants where UAG sites are disrupted (blue curves), measured by gel-shift (see ***Figure 4—figure supplement 1***). $K_d$ estimates for region 1 and region 2 are shown (mean of 2 gel-shifts per sequence). (**F**) Western blot analysis of *Jag1* regulation by Msi: top left panel, *Jag1* expression in *Msi1* overexpression cells and controls in cellular fractions (**T**—total lysate, **C**—cytoplasmic and **N**—nuclear fractions). *Jag1* is translationally repressed upon induction of *Msi1* and detected

*Figure 4. Continued on next page*

*Figure 4. Continued*

only in total and cytoplasmic lysates. hnRNP A1, known to shuttle between the nucleus and the cytoplasm and alpha-Tubulin used as loading controls. (**G**) Increased JAG1 protein levels in double knockout cells. (**H**) Reduced JAG1 protein levels upon *Msi2* overexpression.

The following figure supplements are available for figure 4:

**Figure supplement 1**. Validation by gel-shift of MSI1 binding to Jag1 3' UTR sequences.

**Figure supplement 2**. Effect of *Msi1* gain and loss of function on *Jag1* mRNA levels and protein expression.

**Figure supplement 3**. Validation of Msi-dependent regulation of *Jag1* protein levels using luciferase reporters containing *Jag1* 3' UTR.

to human cells. In sum, our results support a model where Msi proteins directly bind to the *Jag1* 3′ UTR to mediate post-transcriptional repression of protein levels.

## Msi proteins regulate alternative splicing

Since some of the largest changes in translation observed by Ribo-Seq affected RBPs with functions in RNA splicing, we hypothesized that Msi overexpression might trigger changes in pre-mRNA splicing. Changes in mRNA splicing following Msi overexpression or depletion were assessed by analysis of RNA-seq data using the MISO software (*Katz et al., 2010*). For example, exon 38 in the *Myo18a* gene, which is predominantly included under control conditions, was modestly repressed following *Msi2* overexpression and strongly repressed following *Msi1* overexpression (*Figure 5A*). In total, we observed several hundred alternatively spliced exons that were either repressed or enhanced by overexpression or knockout of Msis (*Figure 5B*). Msi proteins are predominantly localized in the cytoplasm (*Figure 5—figure supplement 1*), even when overexpressed (*Figure 3F*), suggesting that these changes in pre-mRNA splicing are indirect. For example, these splicing changes may result from changes in the levels of splicing factors whose mRNAs are translationally regulated by Msi proteins.

To test whether *Msi1* and *Msi2* affect pre-mRNA splicing in similar ways, we compared the direction of splicing changes following *Msi1* or *Msi2* overexpression. Exons with increased inclusion following *Msi1* overexpression tended to show increased inclusion following *Msi2* overexpression as well, while *Msi1* OE-induced splicing changes were uncorrelated with Dox-induced changes (*Figure 5C*). A similar pattern was observed for exons with decreased inclusion (*Figure 5C*). These observations suggested that *Msi1* and *Msi2* trigger similar effects on mRNA splicing. Splicing changes observed in the *Msi1/Msi2* double knockout cells exposed to 4-OHT were inversely correlated to those observed following Msi overexpression (*Figure 5C*). This observation further supports that Msi proteins affect splicing at physiological expression levels. No correlation in splicing was observed between *Msi1*-induced cells and exposure to 4-OHT of double floxed cells lacking the Cre driver (*Figure 5C*).

## Msi-associated splicing changes are observed in cancer lines and associated with luminal state

We next considered whether the splicing changes associated with Msi mis-expression in NSCs might be related to splicing changes observed in human breast cancer cells or with a particular cell state. The natural variation in Msi levels across breast cancer cell lines (*Figure 2C–E*) enabled a comparison of splicing patterns between Msi-high (luminal) vs Msi-low (basal) cells. To compare mouse and human splicing patterns, we identified human alternative exon trios orthologous to mouse alternative and flanking exon trios using synteny in a multi-genome alignment (*Figure 5D* and Supp. 'Materials and methods'). We first compared changes ($\Delta\Psi$) in the percent spliced in (PSI or $\Psi$) values of mouse exons between *Msi1* overexpressing cells vs controls, to $\Delta\Psi$ values of orthologous exons between luminal and basal breast cancer cell lines (*Figure 5E*). The splicing patterns were consistent: the human orthologs of exons up-regulated in Msi1-OE NSCs had higher inclusion in luminal (Msi-high) than in basal (Msi-low) cell lines, and similarly for down-regulated exons (*Figure 5E*). Such agreement was observed for several different luminal and basal pairs, but was strongest when comparing HER2+ luminal lines such as BT474 and SKBR3 to basal lines, consistent with the higher Msi levels observed in HER2+ cell lines (*Figure 2D*). These observations support the proposition that Msi contributes to a luminal splicing program in human breast cancers by triggering changes similar to those induced in mouse NSCs.

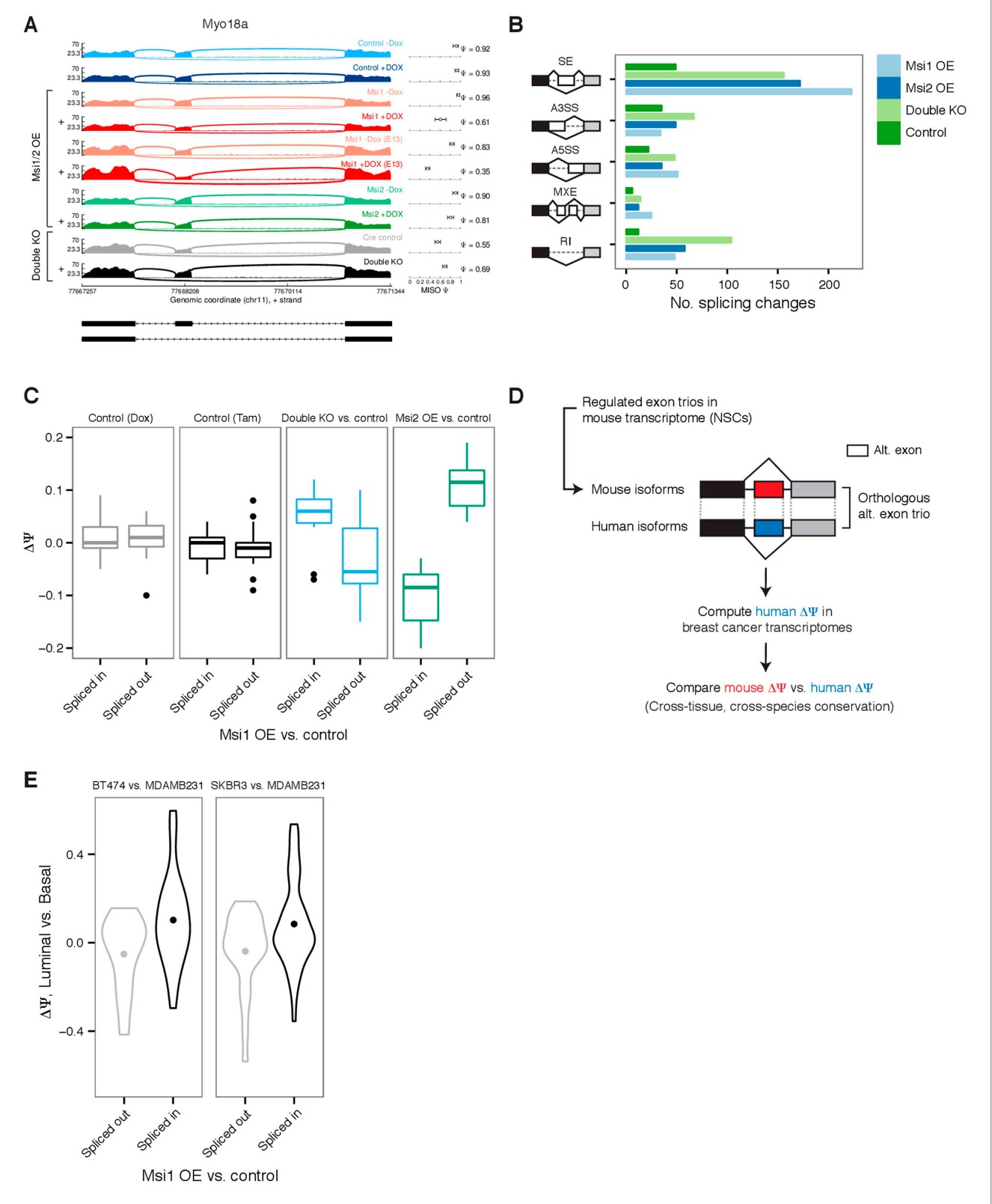

**Figure 5**. Global impact of Msi proteins on alternative splicing. (**A**) Sashimi plot for *Myo18a* alternative exon 38 with Percent Spliced In (Ψ) estimates by MISO (values with 95% confidence intervals, right panel.) Exon splicing is repressed by *Msi1* overexpression and slightly increased in knockout *Msi1/2* cells. '+' indicates samples treated with Dox/Tam for overexpression/knockout cells, respectively. E12.5 neural stem cells were used for all samples

*Figure 5. Continued on next page*

*Figure 5. Continued*

except *Msi1* overexpression for which an additional E13.5 NSC time point was sequenced. (**B**) Number of differential events (MISO Bayes factor ≥10, ΔΨ ≥ 0.12) in each alternative RNA processing category (SE—skipped exons, A5SS—alternative 5′ splice site, A3SS—alternative 3′ splice site, MXE—mutually exclusive exons, RI—retained introns) for *Msi1* overexpression ('Msi1 OE'), *Msi2* overexpression ('Msi2 OE'), double knockouts ('Double KO'), and a Dox control pair ('Control'). (**C**) Comparison of ΔΨ in *Msi1* overexpression vs control binned by direction ('Spliced in' or 'Spliced out', x-axis) to ΔΨ in *Msi2* overexpression cells and in double knockout cells (along with respective Tam and Dox controls, y-axis). (**D**) Computational strategy for identifying human orthologs of alternative exon trios regulated in mouse neural stem cells. Orthologous exon trios were identified by synteny using multiple genome alignments. (**E**) Comparison of ΔΨ mouse alternative exons by *Msi1* (comparing overexpression to control, x-axis) and ΔΨ of their orthologous exon trios in human (comparing luminal and basal cell lines, y-axis). Two pairs of luminal and basal cells compared: BT474 vs MDAMB231 and SKBR3 vs MDAMB231. ΔΨ value distributions summarized by violin plots with a dot indicating the mean ΔΨ value.

The following figure supplements are available for figure 5:

**Figure supplement 1**. Subcellular localization of MSI1 protein in murine NSCs.

**Figure supplement 2**. Analysis of two conserved Msi-induced splicing changes in breast cancer tumors.

Two of the most strongly affected alternative exons in murine NSCs, *Myo18a* exon 38 (*Figure 5A*) and *Erbin* exon 21 (Erbb2ip, a direct binding-partner of the breast cancer oncogene HER2/Erbb2) were conserved in the human genome and detected in the transcriptomes of all analyzed breast tumors and controls. In primary tumors, these exons showed a striking cancer-associated splicing pattern, with the *ERBIN* exon enhanced in tumors and the *MYO18A* exon repressed in tumors (*Figure 5— figure supplement 2A*). To test whether the regulation of these exons is responsive to Msi levels, we correlated the fold change in Msi expression for each matched tumor–control pair with the ΔΨ value of the *ERBIN* and *MYO18A* exons in that pair (*Figure 5—figure supplement 2B*). We observed high correlation between the extent of Msi overexpression and the change in splicing in luminal tumors, particularly for *MSI2*. As in mouse NSCs, increased expression of Msis was associated with increased inclusion of the *ERBIN* exon and repression of *MYO18A* exon splicing, suggesting that Msi-dependent regulation of splicing may be conserved not only in breast cancer cell lines but also in primary tumors.

## Msi proteins are required to maintain epithelial-luminal state in breast cancer cells and regulate EMT processes

To address whether Msi proteins are functionally required for the maintenance of the luminal state, we performed RNAi knockdown of *Msi1* and *Msi2* in two luminal breast cancer cell lines, BT474 and MCF7-Ras, where Msi proteins are highly expressed (*Figure 2C* and *Figure 6—figure supplement 1A*). In the HER2+ luminal cell line BT474, cells grow in tightly packed epithelial colonies (*Figure 6A*). We observed a striking morphological change upon knockdown of *MSI1* or *MSI2*, where cells progressively separated and acquired a basal-like appearance 3–5 days after knockdown (*Figure 6A*), accompanied by reduced proliferation (not shown). A similar phenotype was observed in MCF7-Ras cells upon knockdown of *MSI1* or *MSI2* (*Figure 6—figure supplement 1B*). These results argue that Msi expression is required for the maintenance of the epithelial-luminal state in breast cancer cell lines.

The Notch pathway regulator *Jag1*, which we found was translationally repressed by Msi, is known to be required for EMT. *Jag1*-depleted keratinocytes undergoing TGFβ-induced EMT fail to express mesenchymal markers and retain epithelial morphology (*Zavadil et al., 2004*). Furthermore, knockdown of *Jag1* in keratinocytes strongly impairs wound healing (*Chigurupati et al., 2007*), a process that requires cells to acquire mesenchymal properties such as migration and protrusion. Our gene expression analysis also supported the mesenchymal-basal specific expression of *Jag1*, which is particularly pronounced in breast cancer (*Figure 2*). The epithelial-associated expression pattern of Msi genes and the antagonistic relation between Msi and *Jag1* (*Figure 2*) prompted the hypothesis that Msi activation promotes an epithelial cell identity, effectively blocking EMT.

To test the hypothesis that Msi activation may hinder EMT processes by promoting the epithelial state, we assessed the effect of Msi knockdown and overexpression on EMT marker expression. Knockdown of *MSI1* or *MSI2* in the luminal cell line BT474 generally resulted in a decrease in epithelial marker expression and an increase in mesenchymal marker expression, consistent with Msi loss promoting EMT (*Figure 6B*). To test whether ectopic expression of Msi in mesenchymal cancer cells can

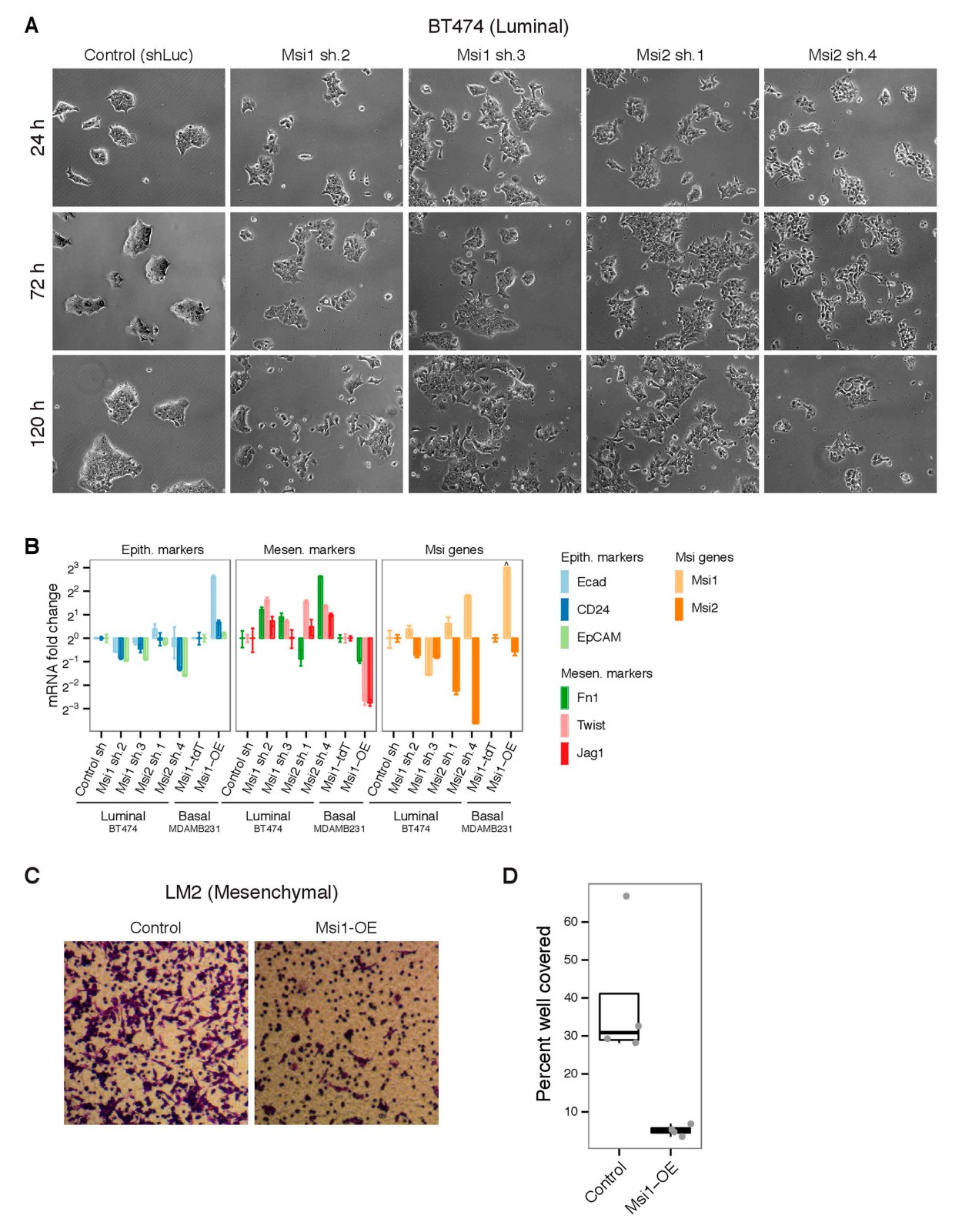

**Figure 6**. Msi levels alter EMT processes breast cancer cell lines. (**A**) Knockdown of *Msi1/Msi2* in BT474 breast cancer cell line using lentiviruses carrying short hairpins (shRNAs). Brightfield images (10x magnification) shown at 24, 72, and 120 hr after Puromycin-selection. (**B**) mRNA expression of epithelial and mesenchymal markers upon knockdown of *Msi1/Msi2* in epithelial-luminal breast cancer cell line (BT474) and overexpression of *Msi1* in

*Figure 6. Continued on next page*

*Figure 6. Continued*

mesenchymal-basal line (MDAMB231). Values plotted are fold changes normalized to GAPDH. For BT474 knockdown, cells infected with hairpin against luciferase were used as control ('Control sh'). For MDAMB231 overexpression, cells infected with tdTomato were used as controls ('Msi1-tdT'). *Msi1* levels were below detection limit in control MDAMB231 cells, therefore *Msi1* fold change in MDAMB231 *Msi1*-overexpression cells (relative to controls) was truncated arbitrarily in plot, indicated by '^'. (**C**) Representative transwell assay image for LM2 control and Msi1-OE breast cancer cells. (**D**) Quantification of percent of well covered in transwell assay for LM2 control and Msi1-OE cells (4 wells per condition, individual well values plotted as dots.).

The following figure supplement is available for figure 6:

**Figure supplement 1**. Knockdown of *Msi1/2* in breast cancer cell lines.

promote an epithelial state, we overexpressed *Msi1* in the mesenchymal cell line MDAMB231, where *Msi1* levels are extremely low. *Msi1*-overexpressing cells had decreased mesenchymal marker expression and increased levels of epithelial marker expression (*Figure 6B*), consistent with promotion of the epithelial state. We conclude that Msi activation promotes the epithelial state in breast cancer cells.

We next asked whether the increase in epithelial markers following Msi overexpression is accompanied by functional changes that reflect the epithelial state. We predicted that ectopic expression of Msi proteins in a mesenchymal cell line would hinder EMT-associated processes such as migration. *Msi1* overexpression in the LM2 cell line (an MDAMB231-derivative) resulted in sevenfold reduction in migration in a transwell assay (*Figure 6C,D*). We were unable to observe this phenotype in the mesenchymal cell lines MDAMB231 or SUM159, where *Msi1* overexpression caused no significant change in migration in the same transwell assays (data not shown). In NSCs, overexpression of *Msi1* or *Msi2* impaired migration as assayed by a scratch assay as well (data not shown), consistent with the phenotype observed in LM2 breast cancer cells. These results show that depending on the cell-type context, Msi activation can decrease the migration capacity of cells, consistent with promotion of an epithelial state and suppression of mesenchymal properties.

## *Msi2* overexpression in the basal cell layer perturbs mammary ductal branching

The association of Msis with the luminal state in breast cancer tumors and their effect on the epithelial-luminal state in breast cancer cell lines prompted us to ask whether Msi proteins play similar roles in the mammary gland in vivo. During maturation, epithelial cells in the mammary gland migrate and form ducts within the mammary fat pad through a process termed mammary ductal branching morphogenesis. The formation of the mammary ductal system is thought to be a kind of EMT (*Chakrabarti et al., 2012*; *Foubert et al., 2010*), making mammary gland an attractive system to study the regulation of EMT in vivo.

The mammary gland Terminal End Buds (TEBs) from which ducts form are organized into discrete layers of cell types, including epithelial luminal and basal cells. The identity of luminal and basal tumors is thought to resemble their mammary gland cell type counterparts. Analysis of RNA-Seq expression analysis of purified mouse mammary luminal ($CD24^{high}CD29^+$) and basal ($CD24^+CD29^{high}$) cells generated by *dos Santos et al. (2013)* revealed enrichment of *Msi1* and *Msi2* expression in luminal cells (not shown). As predicted by the mRNA expression profile, we observed higher MSI2 protein levels in the luminal cell layer and far lower levels in the basal (K14-positive) cell layer of mouse mammary ducts (*Figure 7A*).

We next examined the effect of Msi overexpression on epithelial cell state in the mammary gland in order to see whether its in vivo effects on epithelial-luminal state are similar to those observed in culture models. We ectopically expressed *Msi2* in the basal cell layer, where it is nearly absent normally (*Figure 7A*), using a basal cell-specific Dox-inducible driver, K14-rtTA. As expected, mice administered Dox showed significantly higher levels of MSI2 protein in the basal cell layer (*Figure 7—figure supplement 1A*) and overall higher levels of *Msi2* mRNA in mammary epithelial cells (*Figure 7B*).

Overexpression of *Msi2* altered mammary ductal branching morphology (*Figure 7C*). Overexpression mice showed both a defective and delayed mammary ductal branching pattern. *Msi2* overexpression resulted in fewer mammary duct branch points given, after either 4 or 7 weeks of induction with Dox, with the difference between controls and overexpression mice more pronounced after 7 weeks (*Figure 7—figure supplement 1B*). The TEBs in glands overexpressing *Msi2* were smaller relative to controls, following either 4 or 7 weeks of induction (*Figure 7C*, right inset). In addition, after 4 weeks

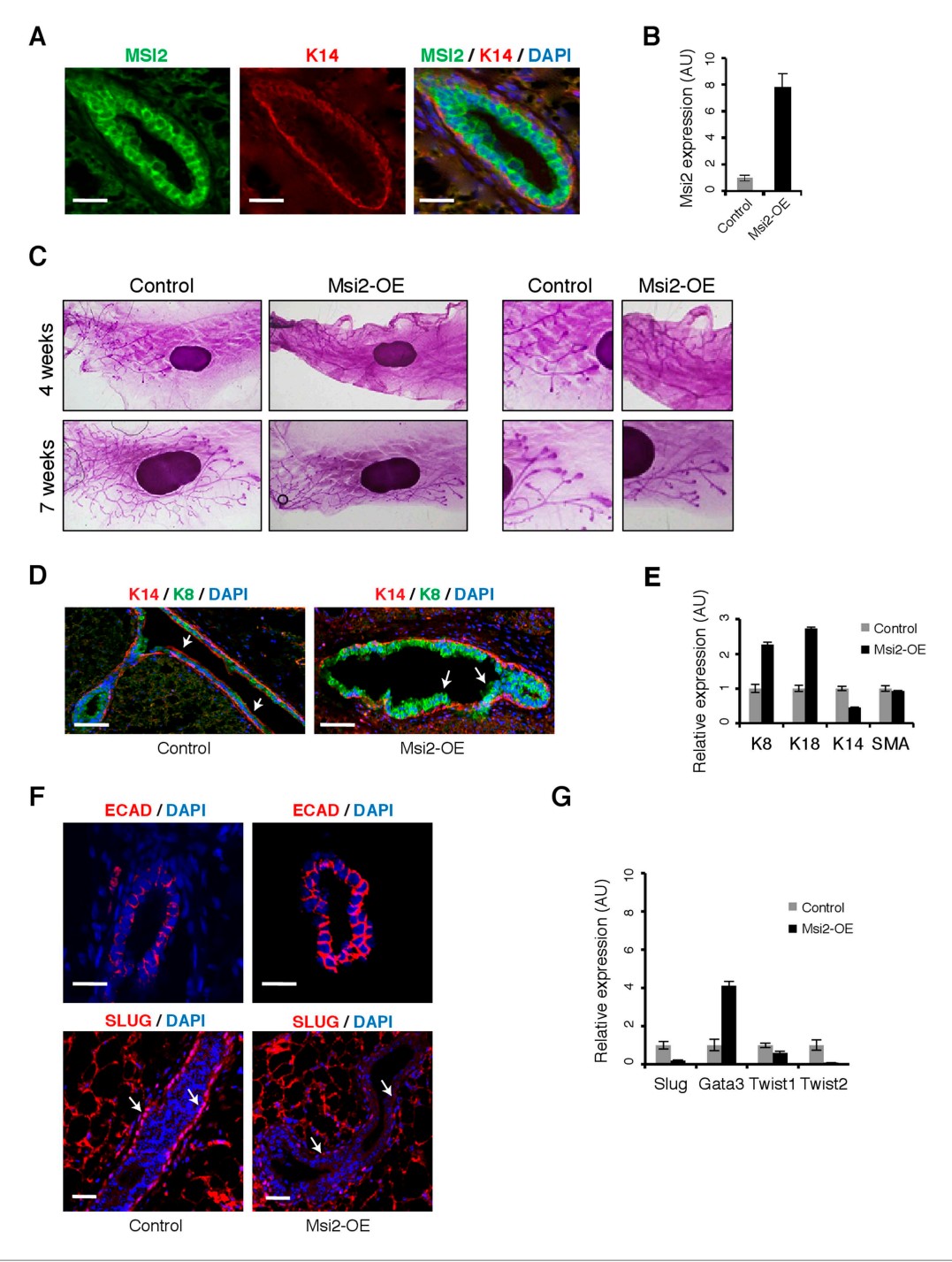

**Figure 7**. Msi2 activation represses EMT and expands mammary luminal cell layer in vivo. (**A**) Immunostaining for MSI2, K14, and DAPI in control sections of mammary gland. Scale bar: 50 µm (**B**) qRT-PCR for *Msi2* in mammary epithelial cells from control and *Msi2* overexpressing mice ('Msi2-OE'). (**C**) Whole mount stain for mammary glands from control and *Msi2* overexpressing mice (left: low magnification, right: high magnification.) (**D**) Immunostaining for K14, K8, and DAPI in mammary gland sections from control and *Msi2* overexpressing mice. Scale bar: 100 µm (**E**) qRT-PCR for luminal markers (K8, K18), basal markers (K14), and smooth-muscle Actin (SMA) in mammary epithelial cells from control and *Msi2* overexpressing mice. (**F**) Staining for E-cadherin (ECAD) (top) and EMT-marker SLUG (bottom) in mammary glands from control and *Msi2* overexpressing mice. Luminal cell layer is expanded upon Dox

*Figure 7. Continued on next page*

*Figure 7. Continued*

(arrows). Scale bar: 100 µm. (**G**) qRT-PCR for Slug, Gata3, Twist1, Twist2 in mammary epithelial cells from control and *Msi2* overexpressing mice. Slug expression in basal cell layer is reduced upon Dox (arrows). Scale bar: 50 µm.

The following figure supplements are available for figure 7:

**Figure supplement 1**. *Msi2* overexpression in mouse mammary gland alters mammary duct morphology.

**Figure supplement 2**. *Msi2* overexpression in mouse mammary gland represses Slug and Jag1.

of induction, glands from overexpression mice had shorter ductal lengths relative to controls, but ductal lengths returned to lengths similar to wild type after 7 weeks of induction (***Figure 7—figure supplement 1C***). These results indicate that *Msi2* overexpression resulted in a defect in mammary branching morphogenesis (evidenced by the reduced number of branch points), and a delay in this process, as indicated by the slower rate of branch ductal growth.

Since branching morphogenesis requires cells to lose their epithelial identity and undergo migration, we hypothesized that the observed defect in branching morphology might result from inability of cells to lose their epithelial identity and/or expansion of an epithelial cell layer. Consistent with this hypothesis, we observed that *Msi2* overexpression resulted in expansion of the luminal cell layer (***Figure 7D*** and ***Figure 7—figure supplement 1D***), confirmed by a corresponding increase in expression of luminal cell markers and a decrease in basal markers (***Figure 7E***). Furthermore, *Msi2* overexpression led to an increase in epithelial marker E-cadherin and reduction in Slug, a marker of EMT and mesenchymal cells. Expression of EMT regulators *Slug*, *Twist1*, and *Twist2* decreased upon *Msi2* overexpression, while expression of the luminal epithelial cell marker *Gata3* increased (***Figure 7G*** and ***Figure 7—figure supplement 2A***). Expression of JAG1 protein was also reduced upon *Msi2* overexpression, consistent with the results observed in murine NSCs (***Figure 7—figure supplement 2B,C***). These results support a model in which ectopic Msi expression leads to expansion of epithelial-luminal cells in the mammary gland, effectively blocking EMT processes required for normal branching morphogenesis, and resulting in the defective ductal branching pattern described above. The observed functions of Msi proteins in regulation of mammary epithelial cell state mirror the functions we observed in breast cancer cell lines and murine NSCs, and suggest that Msi proteins play similar roles in a healthy in vivo context as in cancer cells.

## Discussion

The specific expression patterns of Msi proteins in stem and epithelial cells have aroused interest in their functional roles. Here, we show that Msi proteins are associated with the epithelial-luminal cell state in several cancer types, notably breast cancer, where Msi genes are highly enriched in luminal tumors and luminal breast cancer cell lines. We showed that in breast cancer cells, knockdown of Msi genes leads to loss of epithelial identity and upregulation of mesenchymal markers, while their ectopic activation promotes the epithelial state and suppresses mesenchymal properties such as cell migration. As in cancer cells, overexpression of *Msi2* in healthy mammary gland tissue suppressed EMT and resulted in a defective mammary ductal branching pattern. These observations all support a role for Msi proteins in maintenance of a luminal/epithelial cell state and inhibition of EMT (***Figure 8***). The consistency between our observations in mammary epithelial cells and NSCs and between mouse and human suggests that these functions are shared across cell types and evolutionarily conserved.

Our genome-wide data support the hypothesis that Msi proteins are translational regulators. We showed that Msi proteins can translationally repress *Jag1*, an important regulator of Notch signaling. However, the role of Notch signaling in cancer remains complex and may vary between cancer types (***Dickson et al., 2007***; ***Lobry et al., 2011***). The upregulation of *Jag1* in the basal state suggests that Notch pathway activity is high in and required for the entry into the mesenchymal state, consistent with previous studies (***Zavadil et al., 2004***; ***Dickson et al., 2007***). In mammary epithelial cells, *Jag1*-triggered activation of Notch was shown to reduce E-cadherin expression and increase Slug expression (***Leong et al., 2007***). Furthermore, *Jag1* activation in breast cancer cells promotes their metastasis

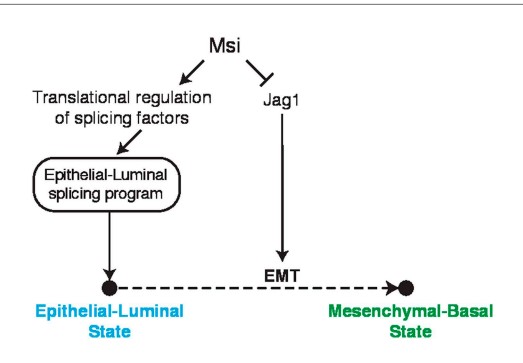

**Figure 8**. Model for Msi roles in regulation of cell state. Model for Msi role in the control of the epithelial state. We show that Msi represses translation of *Jag1*, a positive regulator of Notch and EMT. We also show that Msi promotes expression of an epithelial-luminal splicing program, which we hypothesize occurs through translational regulation of splicing factors. In the model, both the direct regulation of *Jag1* and indirect regulation of splicing contribute to maintenance of an epithelial-luminal cell state and inhibition of EMT.

into the bone in vivo by activating Notch in neighboring bone cells (*Sethi et al., 2011*). The dependence of EMT on Notch activation has been observed in normal development as well. During heart development, cardiac valves are generated from endocardium through EMT, and Notch activity was shown to be required for this process (*Timmerman et al., 2004*). Collectively, these studies are consistent with our working model in which Msi represses *Jag1* translationally, in turn altering Notch activity required for EMT.

The molecular mechanisms by which Msi proteins regulate translation of a subset of mRNAs like *Jag1* remains unclear. Our genome-wide data and in vitro binding assays indicate that Msi proteins act by binding UAG-containing motifs at 3' UTRs of messages. A model where Msi proteins repress translation by outcompeting eIF4G for PolyA-binding protein (PABP) was proposed (*Kawahara et al., 2008*), but the conditions under which binding to mRNA results in translational repression are unclear, since only a subset of mRNAs are detectably regulated. It is possible that co-factors are required in vivo for Msi to affect translation following binding to the mRNA. It is also possible that other RNA-binding factors outcompete Msi protein for binding, though MSI1 has relative high RNA-binding affinity. The molecular mechanism underlying Musashi-dependent translational control and the nature of any co-factors involved are not known.

This study complements recent reports of the involvement of post-transcriptional regulatory factors in cell state maintenance and EMT. For example, the epithelial-specific splicing factors of the ESRP family play important roles in maintenance of epithelial state (*Warzecha et al., 2009*; *Reinke et al., 2012*). A recent study presented evidence that the transcription factor Snail can promote the mesenchymal state in part by repressing *Esrp1* (*Reinke et al., 2012*), further highlighting the importance of post-transcriptional control in driving cell state transitions like EMT.

Like master transcription factors, master post-transcriptional regulatory factors globally alter gene expression—by affecting RNA splicing, stability, localization, or translation—which makes them suitable for controlling cell identity (*Jangi and Sharp, 2014*). Our study shows that post-transcriptional regulatory factors like Msi proteins can impact both translation and pre-mRNA splicing, utilizing multiple layers of RNA regulation to reshape the transcriptome for a particular cell state. Many of the impacted splicing events are part of an epithelial splicing program, suggesting that effects of Msis on splicing may reinforce the effects of *Jag1* repression on maintenance of epithelial cell state. The predominantly cytoplasmic expression of Msis makes it likely that splicing is affected indirectly, e.g., through translational regulation of specific splicing factors, though our data do not rule out that a small fraction of Msi protein may be nuclear localized and could directly regulate splicing. We have also observed that other RBPs are also enriched in the epithelial state (*Shapiro et al., 2011*), suggesting that RBPs as a group may play a broad role in maintenance of this state, and might provide attractive targets for therapeutic efforts to manipulate cell state.

Msi proteins are co-expressed with various proliferation markers in a wide variety of stem cell niches, including the breast, stomach, intestine, lung, and brain. This observation suggests the hypothesis that Msis may act as general epithelial stem cell/progenitor regulators across tissues. Our findings are consistent with this hypothesis, but further study of Msi in multiple stem cell compartments will be needed to directly test it. The role of Msi in the normal development and transformation of other adult tissues will also be important to understand. For example, our observation that Msi is frequently overexpressed in lung tumors suggests that ectopic expression of Msi proteins in the lung could elucidate their role in lung cancer. Furthermore, the systematic downregulation of *Msi1/Msi2* and high frequency of *Msi1* mutations in kidney tumors suggests that kidney would be an informative model for studying Msi loss-of-function and its consequences in cancer.

# Materials and methods

## Mouse strains and derivation of neural stem cell lines

Inducible overexpression mice (tetO-Msi1/Msi2) were generated as previously described in *Beard et al. (2006)*; *Kharas et al. (2010)*. The generation of *Msi2* conditional knockout mice was previously described in *Park et al. (2014)*, and the generation of *Msi1* conditional knockout mice will be described elsewhere (Yu et al., under review). Mice of the 129SvJae strain were used, and the K14-rtTA strain was obtained from JAX (stock number: 007678). Animal care was in accordance with institutional guidelines and approved by the Committee on Animal Care, Department of Comparative Medicine, Massachusetts Institute of Technology, under animal protocol 1013-088-16. For derivation of embryonic neural stem cells (NSCs), littermate embryos were used whenever possible. Cortical NSCs were derived from embryos following *Kim et al. (2003)*. Briefly, cortical tissue was isolated from E12.5 embryos (unless otherwise noted) under a light dissection microscope inside a sterile fume hood and collected by centrifugation. Cortical tissues were dissociated into single cells by trituration in Magnesium/Calcium-free HBSS buffer (Gibco, Woburn MA) followed by 15-min incubation at room temperature. Dissociated tissue was collected by centrifugation, resuspended in N2 medium containing growth factors and Laminin (Life Technologies, Woburn MA, Catalog Number: 23017015) and plated onto Polyornithin/Laminin-coated tissue culture dishes as in *Okabe et al. (1996)*.

## Culture conditions for embryonic neural stem cells

NSCs were grown in N2 medium (*Okabe et al., 1996*) containing EGF (20 ng/ml) and bFGF (20 ng/ml) and Laminin (Life Technologies). Cells were grown on Polyornithin/Laminin-coated dishes. EMT was induced by switching cells to N2 medium containing LIF/FBS as described in *Ber et al. (2012)*.

## Culture conditions for human breast cancer lines, shRNA knockdowns and overexpression assays

All breast cancer lines were cultured in DME containing 10% FBS, 1% GlutaMAX (Gibco), and Penn/Strep, except for BT474, which was cultured in RPMI base medium, and SKBR3 which was cultured with McCoy's 5A supplement. Lentiviruses carrying pLKO vectors with hairpins against *Msi1*, *Msi2*, or Luciferase (control) were used for knockdowns. Hairpins were obtained from Broad Institute shRNA library. Cells were infected in a centrifuge spin-infection step (1500 RPM, 37°C, 20 min) following a 2-hr incubation with polybrene or protamine sulfate, and viral medium was added to the cells overnight. Cells were subjected to 4–6 day Puromycin selection (2 µg/ml) 48 hr after infection. Msi1-OE vector (Thermo OpenBiosystems) was used for overexpression assays. Virus was prepared was described above and cell lines infected with virus were selected for 4–6 days with Blasticidin (5 µg/ml) 48 hr after infection.

## Migration assay in breast cancer cell lines

Migration assay was performed using the transwells (Corning 6.5 mm Diameter inserts with 8um pore size, polycarbonate membrane; product #3422, lot #19614003). 50,000 cells were seeded into wells in each condition and allowed to migrate for 9 hr. Cells were stained with Crystal Violet and then percent area covered was calculated using ImageJ. Images were threshold filtered on Hue and Saturation (Hue: 192-255 'pass'; Saturation: 72-255 'pass') and passed to the 'Analyze Particles' function with a threshold size of 2000.

## Western blotting, immunofluorescence staining, and antibodies used

For western blotting, cells were lysed on ice and protein lysates were loaded onto 4-12% gradient Bis-Tris Gel (Life Technologies). Primary antibodies and dilutions used in western blotting on murine NSCs: anti-MSI1/2 (Cell Signaling Technology #2154, 1:800), anti-MSI2 (Abcam #57341, 1:800), anti-Jag1 (Cell Signaling Technology #2620, 1:800), anti-HER2 (Cell Signaling Technologies #2248, 1:1000), anti-phos-HER2 (Cell Signaling Technology #2241, 1:1000), anti-alpha-Tubulin (Sigma-Aldrich T9026, 1:5000), anti-HNRNPA1 (Abcam ab5832, 1:800). Immunofluorescene was performed on cells grown on glass bottom chambers (LabTek II, #1.5), fixed in 4% PFA. Cells were blocked and permeabilized in 5% FBS, .1% Triton in PBS(+). Antibodies were applied in 1% FBS in PBS(+). Immunofluorescence antibodies and dilutions: anti-MSI1 (MBL D270-3, 1:500), anti-HNRNP A2/B1 (Santa Cruz, sc-374052, 1:200). For IHC on murine mammary glands, anti-Jag1 (Santa Cruz, SC-6011, 1:100) was used. For western on murine mammary glands, anti-Jag1 (Santa Cruz, SC-6011, 1:1000) and anti-Tubulin (Sigma-Aldrich, T5168, 1:4000) were used.

## Immunohistochemistry on human breast cancer sections

Paraffin-embedded human breast cancer sections were obtained from Biomax US (BR1505a) and stained using standard protocols with antigen retrieval. Antibodies used: anti-ECAD1 (BD Biosciences, 1:50) and anti-MSI1 (MBL D270-3, 1:200).

## Confocal imaging for immunofluorescence

Confocal imaging was performed using a Perkin–Elmer microscope using oil-immersion 63× objective, imaged with Velocity software. Single confocal stacks or maximum Z intensity projections were obtained using Fiji (Bioformats-LOCI plugin).

## RNA-seq and ribosome profiling library generation

RNA-Seq libraries were prepared from polyA-selected RNA using standard Illumina protocol. Ribosome profiling libraries were prepared following *Ingolia et al. (2009)* with several modifications. Briefly, cells were collected by centrifugation and immediately flash-frozen. Cells were thawed in lysis buffer (20 mM HEPES [pH 7.0], 100 mM KCl, 5 mM MgCl2, 0.5% Na-Deoxycholate, 0.5% NP-40, 1 mM DTT, Roche mini EDTA-free protease inhibitor tablets [1 tablet/10 ml]) and briefly treated with DNase I and RNAse I. Nuclei and cell debris were removed by centrifugation and lysates were treated with RNase I (NEB) for 75 min at room temperature to generate monosome-protected RNA fragments. Monosomes were collected by ultracentrifugation in a sucrose cushion, denatured in 8 M Guanidium HCl, and protected RNA fragments (footprints) were extracted with Phenol–Chloroform. Footprints were dephosphorylated by PNK treatment and size-selected (~31–35 nt fragments) by purification from a 15% TBE-Urea gel. Subtractive hybridization of ribosomal RNA from footprints was performed as in (*Wang et al., 2012*). Footprints were then polyA-tailed, and Illumina sequencing adaptors were added in a reverse transcription step to obtain footprint cDNA, which was then isolated by gel purification. cDNA was then circularized, PCR-amplified, and PCR products isolated by gel purification and submitted for sequencing on Illumina Hi-Seq platform.

## Computational analysis of RNA-Seq, ribosome profiling and bind-n-seq

Source code for the pipelines used to analyze RNA-Seq, ribosome profiling and Bind-n-Seq data is available through the open-source library rnaseqlib (available at the git repository: http://www.github.com/yarden/rnaseqlib). Protocols, raw sequencing data and additional information about genomic datasets are available at http://www.musashi-genes.org.

### Ribosome profiling (ribo-seq) analysis

To define a set of translationally regulated targets, we first filtered out genes that had low read counts (5 reads or less) in constitutive CDS exons in either RNA-Seq or Ribo-Seq data. We then further filtered out from this set genes that showed 1.5-fold change or greater in mRNA levels between control and experimental samples, to avoid instances where changes in TE may be confounded by changes in mRNA abundances, and therefore are less likely to be controlled solely at the level of translation. From this set of genes, we defined the subset that had a threefold or higher change in TE as the set of translational targets.

### Bind-n-seq (RBNS) analysis

To define a set of genes with enriched Msi binding sites, we ranked genes according to the abundance of RBNS-enriched 6mers in their 3' UTR. For each gene $g$, we calculated the density an RBNS-enriched 6mer $k$ in the gene, $D_{g,k,}$ as follows:

$$D_{g,k} = \frac{n_k}{u - 6 + 1}$$

where $n_k$ is the number of occurrences of the 6mer $k$ in the longest 3' UTR of $g$, and $u$ is the UTR length. We defined the enrichment density score $S_g$ for each gene $g$ as the sum of densities of all RBNS-enriched 6mers in the gene:

$$S_g = \sum_k D_{g,k}$$

We then calculated the distribution of $S_g$ for all genes and ranked each gene by its percentile rank. The score for *Jag1* ($S_{Jag1}$) ranked in the 85th percentile of the score distribution.

## On *Numb* as a translational target of Msi proteins

Early work on mammalian Musashi proteins by the Okano group and colleagues suggested that *Numb* mRNA is translationally repressed by MSI1 (*Okano et al., 2002*). A later study by the same group showed that in the gastric system, *Msi1* KO mice had lower, not higher, levels of Numb protein, opposite of the expected change under the translational repression model (*Takahashi et al., 2013*). Recent work in HSCs (where only *Msi2* is expressed) showed a *Numb*-independent phenotype for *Msi2* and found that *Msi2* KO HSCs have unchanged levels of Numb protein (*Park et al., 2014*). Thus, it is unclear if *Msi1* or *Msi2* directly regulate *Numb* mRNA translation in all systems and whether such regulation always promotes or represses translation of the mRNA.

In our data from NSCs, we were unable to detect a large difference in *Numb* translational efficiency upon *Msi1* overexpression as measured by Ribo-Seq, though a small effect cannot be excluded since coverage of the *Numb* mRNA in our Ribo-Seq data was low. It is possible that *Msi1* affects the translation of certain *Numb* mRNA isoforms in a context-specific manner, potentially through alternative mRNA processing of the *Numb* mRNA, as proposed by *Takahashi et al. (2013)*.

## Sequencing data availability

All RNA sequencing data was submitted to GEO (accession GSE58423).

## Computational analysis of TCGA data

Publicly available TCGA data sets (Level 2 and Level 3) were downloaded from NIH 'Bulk Download' website (RNASeqV2: https://wiki.nci.nih.gov/display/TCGA/RNASeq+Version+2). RNA-Seq analyses were performed using 'RNASeqV2' TCGA files. Fold changes for genes were normalized by correction with Lowess-fit of MA-values calculated using raw gene expression estimates. Alternative exon expression was quantified using MISO.

## Computational identification of orthologous exon trios between mouse and human

Syntenic regions for exons in mouse alternative exon trios (mm9) were computed using Ensembl Compara Database (Release 66) PECAN multiple genomes alignment, using the Pycogent Python framework (*Knight et al., 2007*). Syntenic coordinates in human genome (hg19) were then matched to annotated hg19 exon coordinates given in TCGA data files.

## RNA bind-n-seq protein expression, RNA preparation and binding

A streptavidin binding peptide (SBP) tag was added to the pGEX6P-1 vector (GE) after the Presceission protease site. Full-length Musashi (*Msi1*) was cloned downstream of the SBP tag with infusion (Clontech) using BamHI and NotI cloning sites. Expression of tagged MSI1 was induced with 0.5 mM IPTG at 18° for 4 hr in the Rosetta(DE3)pLysS *E. coli* strain and subsequently purified on a GST GraviTrap column (GE). MSI1 was eluted from the GST column with PreScission protease (GE) in 4 mL of Protease Buffer (50 mM Tris pH 7.0, 150 mM NaCl, 1 mM EDTA, 1 mM DTT) at 4° C overnight (~16 hr). Protein purity was assayed SDS-PAGE gel electrophoresis and visualized with SimplyBlue SafeStain (Invitrogen).

Input random RNA was generated by T7 in vitro transcription: 1 µg T7 oligo was annealed to 1 µg of RBNS T7 template by heating the mixture at 65° C for 5 min then allowing the reaction to cool at room temperature for 2 min. The random RNA was then in vitro transcribed with HiScribe T7 In vitro transcription kit (NEB) according to manufacturer's instructions. The RNA was then gel-purified from a 6% TBE-urea gel.

Nine concentrations of purified MSI1 (0 nM, 0.5 nM, 2 nM, 8 nM, 16 nM, 64 nM, 256 nM, 1 µM, and 2 µM) were equilibrated in 250 µl of Binding Buffer (25 mM Tris pH 7.5, 150 mM KCl, 3 mM MgCl2, 0.01% Tween, 1 mg/ml BSA, 1 mM DTT, 30 µg/ml poly I/C [Sigma]) for 30 min at room temperature. 40 U of Superasin (Ambion) and 1 µM random RNA (final concentration) was added to the MSI1 solutions and incubated for 1 hr at room temperature. During this incubation, Streptavidin magnetic beads (Invitrogen) were washed three times with 1 ml of wash buffer (25 mM Tris pH 7.5, 150 mM KCl, 60 µg/ml BSA, 0.5 mM EDTA, 0.01% Tween) and then equilibrated in Binding Buffer until needed. MSI1 and interacting RNA was pulled down by adding the RNA/protein solutions to 1 mg of washed streptavidin magnetic beads and incubated for 1 hr at room temperature. Supernatant (unbound RNA) was removed from the beads and the beads washed once with 1 ml of

Wash Buffer. The beads were incubated at 70° for 10 min in 100 µl of Elution Buffer (10 mM tris pH 7.0, 1 mM EDTA, 1% SDS) and the supernatant was collected. Bound RNA was extracted from the eluate by phenol/chloroform extraction and ethanol precipitation. Half of the extracted RNA from each condition was reverse transcribed into cDNA using Superscript III (Invitrogen) according to manufacturer's instructions using the RBNS RT primer. To control for any nucleotide biases in the input random library, 0.5 pmol of the RBNS input RNA pool was also reverse transcribed and Illumina sequencing library prep followed by 8–10 cycles of PCR using High Fidelity Phusion (NEB). As Msi1 concentration was increased, decreasing input RT reaction was required in the PCR. For instance, the highest MSI1 condition required 30-fold less input RT product than the no MSI1 condition. All libraries were barcoded in the PCR step, pooled together, and sequenced one HiSeq 2000 lane.

## Primers and sequences related to RNA Bind-n-Seq

RBNS T7 template:
CCTTGACACCCGAGAATTCCA(N$_{40}$)GATCGTCGGACTGTAGAACTCCCTATAGTGAGTCGTATTA

T7 oligo:
TAATACGACTCACTATAGGG

Resulting RNA Pool:
GAGTTCTACAGTCCGACGATC(N)40TGGAATTCTCGGGTGTCAAGG

Binding site used for validation:
GGCUUCUUAAGCGUUAGUUAUUUAGUUCGUUUGUU

RBNS RT primer:
GCCTTGGCACCCGAGAATTCCA

RNA PCR (RP1):
AATGATACGGCGACCACCGAGATCTACACGTTCAGAGTTCTACAGTCCGACGATC

Barcoded Primers:
CAAGCAGAAGACGGCATACGAGAT–BARCODE-GTGACTGGAGTTCCTTGGCACCCGAGAATTCCA

*Jag1* region 1 sequence:
UGUCCAGU**UAG**AUCACUGUU**UAG**AU

*Jag1* region 1 mutant:
UGUCCAGU**UCC**AUCACUGUU**UCC**AU

*Jag1* region 2 sequence:
UCAAAG**UAG**AAUUUUUGUA**UAG**UUAUGUAAAUAAU

*Jag1* region 2 mutant:
UCAAAG**UCC**AAUUUUUGUA**UCC**UUAUGUAAAUAAU

## Luciferase reporter assays for protein translation

The *Jag1* 3' UTR was cloned into the pRL-SV40 vector (Promega) downstream of Renilla luciferase using the XbaI and NotI restriction sites creating the Renilla-Jag1-UTR construct. Firefly luciferase expression was used as the internal control and expressed from the PGL3 vector (Promega). Renilla and the Firefly luciferase vectors were co-transfected into 293 cells stably expressing hairpins against *Msi1*, *Msi2*, or both *Msi1* and *Msi2*, or into mock transfected 293T cells. Cells were harvested between 30–36 hr after transfection and the Renilla and Firefly luciferase signals measured using the Dual-luciferase Reporter Assay System (Promega) according to manufacture's instructions.

## In vivo overexpression and whole mount mammary gland staining

Mice were given Dox (Sigma) via drinking water at 2 g/l. Mice were induced with Dox for 7 weeks unless otherwise indicated. Inguinal mammary glands were spread on glass slides, fixed in Carnoy's fixative (6:3:1, 100% ethanol: chloroform: glacial acetic acid) for 2 to 4 hr at room temperature, washed

in 70% ethanol for 15 min, rinsed through graded alcohol followed by distilled water for 5 min, then stained in carmine alum overnight, washed in 70%, 95%, 100% ethanol for 15 min each, cleared in xylene, and mounted with Permount.

## Immunofluorescence on mammary gland sections

Mammary glands were fixed in 4% PFA, paraffin-embedded and 5-μm sections were used for immuno-fluorescence assay. Paraffin sections were microwave pretreated and incubated with primary antibodies, then incubated with secondary antibodies (Invitrogen) and counterstained with DAPI in mounting media. The following antibodies were used: anti-K14 (Abcam), anti-K8 (Abcam), anti-E-cadherin (CST), anti-Msi2 (Novus Biologicals), anti-Hes1 (Abcam), anti-Slug (CST).

## Quantitative RT-PCR analysis in mammary glands

Mouse mammary epithelial cells were prepared according to the manufacturer's protocol (Stem Cell Technologies, Vancouver, Canada). Briefly, following removal of the lymph node, mammary glands dissected from 10-week-old virgin female mice were digested in EpiCult-B with 5% fetal bovine serum (FBS), 300 U/ml collagenase, and 100 U/ml hyaluronidase for 8 hr at 37°C. After vortexing and lysis of the red blood cells in $NH_4Cl$, mammary epithelial cells were obtained by sequential dissociation of the fragments by gentle pipetting for 1–2 min in 0.25% trypsin, and 2 min in 5 mg/ml dispase plus 0.1 mg/ml DNase I (DNase; Sigma). Total RNA was isolated from mammary epithelial cells. Complementary DNA was prepared using the MMLV cDNA synthesis kit (Promega). Quantitative RT-PCR was performed using the SYBR-green detection system (Roche). Primers were as follows:

*Msi2* forward primer: ACGACTCCCAGCACGACC; *Msi2* reverse primer: GCCAGCTCAGTCCACCGATA.

*K8* forward primer: ATCAAGAAGGATGTGGACGAA; *K8* Reverse primer: TTGGCAATGTCCTCGTACTG.

*K14* forward primer: CAGCCCCTACTTCAAGACCA; *K14* Reverse primer: AATCTGCAGGAGGACATTGG.

K18 forward primer: TGCCGCCGATGACTTTAGA; K18 Reverse primer: TTGCTGAGGTCCTGAGATTTG.

## Quantitative RT-PCR analysis in breast cancer cell lines

RNA was extracted using Trizol and cDNA was prepared using SuperScript III (Invitrogen). Primers used are listed below ('h' prefix denotes human gene, 'F' denotes forward primer, 'R' denotes reverse primer):

hEcad-F:
TGCCCAGAAAATGAAAAAGG
hEcad-R:
GTGTATGTGGCAATGCGTTC
hTwist-F:
GGAGTCCGCAGTCTTACGAG
hTwist-R:
TCTGGAGGACCTGGTAGAGG
hEpCAM-F:
CTTTAAGGCCAAGCAGTGCA
hEpCAM-R:
CGCGTTGTGATCTCCTTCTG
hCD24-F:
GGTTTGACTAGATGATGGATGCC
hCD24-R:
TCCATTCCACAATCCCATCCT
hMsi1-F:
GGGACTCAGTTGGCAGACTAC
hMsi1-R:
CTGGTCCATGAAAGTGACGAA
hMsi2-F:

ACCTCACCAGATAGCCTTAGAG
hMsi2-R:
AGCGTTTCGTAGTGGGATCTC
hJag1-F:
GTCCATGCAGAACGTGAACG
hJag1-R:
GCGGGACTGATACTCCTTGA

## Acknowledgements

We thank V Butty, P Reddien, P Sharp, F Soldner, J Muffat, R Weinberg, L Surface, N Spies, R Friedman, M Kharas and M Lodato for helpful discussions, R Flannery for assistance with mouse colony maintenance, and D. Fu for assistance processing histology sections. We thank Shmulik Motola and Stuart Levine (MIT BioMicroCenter) for high-throughput sequencing, and Wendy Solomon from Keck Microscopy Facility (Whitehead Institute) for assistance with microscopy. Supported by NIH grants R01-GM096193 (EMA), RO1-CA084198 (RJ), U01-CA184897 and R01-GM085319 (CBB). ZY and FL are supported by the National Basic Research program of China (973 program, 2011CB944103), the National Natural Science Foundation of China (NSFC, 31271584), and the National Transgenic Breeding Project of China (2011ZX08009-001-003). EMA is supported by Alfred P Sloan fellowship, and ESS by an NSF Graduate Research Fellowship (Grant No. 1122374).

## Additional information

### Competing interests

EMA: Reviewing Editor, *eLife*. The other authors declare that no competing interests exist.

### Funding

| Funder | Grant reference number | Author |
|---|---|---|
| National Institute of General Medical Sciences | R01-GM085319 | Christopher B Burge |
| National Cancer Institute | U01-CA184897 | Christopher B Burge |
| National Cancer Institute | RO1-CA084198 | Rudolf Jaenisch |
| National Institute of General Medical Sciences | R01-GM096193 | Edoardo M Airoldi |

The funders had no role in study design, data collection and interpretation, or the decision to submit the work for publication.

### Author contributions

YK, Conception and design, Acquisition of data, Analysis and interpretation of data, Drafting or revising the article; FL, NJL, Acquisition of data, Analysis and interpretation of data, Drafting or revising the article; ESS, W-LT, Acquisition of data, Drafting or revising the article; AWC, EMA, CJL, Analysis and interpretation of data, Drafting or revising the article, Contributed unpublished essential data or reagents; PBG, ZY, RJ, CBB, Analysis and interpretation of data, Drafting or revising the article

### Ethics

Animal experimentation: Mice of the 129SvJae strain were used, and the K14-rtTA strain were obtained from JAX (stock number: 007678). Animal care was performed in accordance with institutional guidelines and approved by the Committee on Animal Care, Department of Comparative Medicine, Massachusetts Institute of Technology, under animal protocol 1013-088-16.

## Additional files

### Supplementary file

• Supplementary file 1. Breast cancer RNA-Seq datasets used in analysis (apart from TCGA).

## Major dataset

The following dataset was generated:

| Author(s) | Year | Dataset title | Dataset ID and/or URL | Database, license, and accessibility information |
|---|---|---|---|---|
| Katz Y, Burge CB | 2014 | Transcriptome and translatome analysis of Msi in Mouse Neural Stem Cells | http://www.ncbi.nlm.nih.gov/geo/query/acc.cgi?acc=GSE58423 | Publicly available at NCBI Gene Expression Omnibus. |

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
