## [Decision Letter]

Thank you for sending your work entitled “Musashi Proteins are
Post-transcriptional Regulators of the Epithelial-luminal Cell State” for
consideration at *eLife.* Your article has been favorably evaluated by
James Manley (Senior editor), a Reviewing editor, and 2 reviewers.

The Reviewing editor and the two reviewers discussed their comments before we reached
this decision, and the Reviewing editor has assembled the following comments to help you
prepare a revised submission.

This manuscript addresses the function of Msi proteins, a class of RNA binding proteins
for which little is known. The authors initially analyze tumour and normal tissue
RNA-Seq data from the Cancer Genomic Atlas repository, as well as from cancer cell
lines, to document changes between Msi expression and specific cancer/normal cell types.
Ribosome profiling and RNA-sequencing analyses are then used to identify Msi mRNA
targets. The results suggest that Msi functions in establishing epithelial status. An
important finding is that Msi proteins inhibit the translation of the Notch ligand Jag,
which plays an important role in EMT. An indirect role for Msi proteins in alternative
splicing is also suggested. The data supporting the first set of observations and
associated conclusions in the manuscript are extensive, well presented, and backed by
statistical analyses. The second part of the manuscript documents cellular phenotypes
associated with altered expression of Msi. While these data are less quantitative,
overall they are consistent with the view that Msi contributes to establishing and
maintaining the epithelial state in both cancerous and normal developmental
contexts.

Main points:

1) Figure 1. The relationship between Msi
expression and cancer is complex, as expression levels for these factors display
increases and decreases within the same type of cancer, between different cancers, and
there are also substantial differences between Msi1 and Msi2 expression. Given that
tumors are typically highly heterogeneous and that epithelial tumor tissues are often
contaminated with surrounding normal stromal/mesenchymal cells, do the above variations
reflect this heterogeneity? While the subsequent analysis of cell lines partially
addresses this issue, the authors should at least discuss that levels of Msi in tumour
tissues may be more reflective of epithelial content than cancerous state.

2) It would be informative to use the Cancer Genomic Atlas data to compare expression
levels of additional RBPs that have been linked to post-transcriptional regulatory
programs associated with EMT/MET transitions, such as RBFOX2 and MBNLs. In this regard,
the authors are referred to relevant work by Venables et al. (Mol Cell Biol. 2013
Jan;33(2):396-405), which should be referenced.

3) The authors' study would be strengthened by providing a more definitive
functional link between one or more targets of Msi1/2 and epithelial state. For example,
they could test whether Jag1 knockdown and/or over-expression rescues Msi manipulations
in scratch wound (see below), or cell scatter assays in Figure 6, respectively.

4) The authors should provide more information on how they define Msi translation
targets. How many genes were considered as targets? What fraction of genes have 3'
UTRs enriched in UAGs (in the 85% percentile range) that are not regulated by Msi?

5) The changes in splicing are proposed to be indirect, in large part because the bulk
of Msi proteins are cytoplasmic. However, a possible direct role should be acknowledged
in the absence of additional data.

6) It is an overstatement to say that Msi OE inhibits wound healing. Wound healing is a
complex process that is tightly regulated and involves multiple layers of tissues acting
in a coordinated way. To say that Msis inhibit wound healing may be taken to suggest
that the authors have observed Msis actually acting in the process of wound healing
rather than in the in vitro phenotypic scratch wound assay. The authors should change
their language from “wound healing” to “migration”.

7) The authors should verify induction of EMT upon knock down of Msis in the cell
scatter assays in Figure 6, and recovery of an
epithelial phenotype upon Msi expression in the scratch assays.

8) Msi OE causes a slight delay in cell migration as it may also do to mammary ductal
branching. The defect in mammary ductal branching is actually unclear (is it amplitude,
number of branches?). Where possible, quantification should be applied. The authors
propose that mammary gland development is a « a type of EMT ». References
linking mammary development to EMT should be provided.

9) Typically, EMT has been linked to metastasis. It would be most relevant to test the
impact of Msi OE on the metastatic potential of cancer cells injected into mice.

---

## [Author Response]

Our primary new results are summarized below. We have:

1) Showed that knockdown of Msi1/Msi2 in luminal breast cancer cell lines decreases
epithelial marker expression, and increases mesenchymal marker expression. We showed
this using qRT-PCR for EMT markers, as suggested by reviewers. We additionally
overexpressed Msi1 in a mesenchymal breast cancer cell line, which resulted in reduced
mesenchymal marker expression and increased epithelial marker expression, supporting our
model that Msi activation promotes an epithelial state.

2) Performed a transwell migration in a mesenchymal cancer cell line and showed that
overexpression of Msi1 significantly hinders migration, consistent with Msi activation
suppressing EMT.

3) Further explored regulation of Msi targets, by performing luciferase reporter assays
for the Notch-ligand *Jag1.* We showed that knockdown of Msi1/Msi2 in
293T cells increases expression of the *Jag1* 3’ UTR reporter,
supporting our model that *Jag1* protein expression is regulated by Msi
binding to its 3' UTR. This result not only further strengthens Msi’s role
in regulating *Jag1*, but also demonstrates that this regulation is
conserved in human cells. Finally, we showed that Msi2 overexpression in the healthy
mouse mammary gland results in reduced *Jag1* expression. Together, our
results show that Msi proteins regulate *Jag1* in mouse and human, and
across distinct cell types (NSCs and mammary epithelial cells).

4) Extended our computational analyses to other EMT-associated RNA-binding proteins in
the TCGA dataset, as suggested by reviewers, and clarified our methods of analysis.

5) Quantified the ductal branching phenotype in mammary glands of Msi2 overexpressing
mice.

Main points:

*1)*
Figure 1*. The
relationship between Msi expression and cancer is complex, as expression levels for
these factors display increases and decreases within the same type of cancer, between
different cancers, and there are also substantial differences between Msi1 and Msi2
expression. Given that tumors are typically highly heterogeneous and that epithelial
tumor tissues are often contaminated with surrounding normal stromal/mesenchymal
cells, do the above variations reflect this heterogeneity? While the subsequent
analysis of cell lines partially addresses this issue, the authors should at least
discuss that levels of Msi in tumour tissues may be more reflective of epithelial
content than cancerous state*.

We agree that tumor heterogeneity is an important issue that should be discussed and we
have added text discussing the possibility that increased Musashi levels may reflect the
higher content of epithelial cells in certain tumors.

*2) It would be informative to use the Cancer Genomic Atlas data to compare
expression levels of additional RBPs that have been linked to post-transcriptional
regulatory programs associated with EMT/MET transitions, such as RBFOX2 and MBNLs. In
this regard, the authors are referred to relevant work by Venables et al. (Mol Cell
Biol. 2013 Jan;33(2):396-405), which should be referenced*.

We examined the expression of RBFOX2 and MBNL1 in breast cancer tumors from TCGA.
Consistent with the findings of Venables et. al. (2013), we observed that both RBFOX2
and MBNL1 are more highly expressed in basal tumors compared with the epithelial-luminal
tumor subtypes. These data are shown in Figure 2—figure supplement 2, and we have added a citation of Venables et. al.
2013 to the text.

*3) The authors' study would be strengthened by providing a more definitive
functional link between one or more targets of Msi1/2 and epithelial state. For
example, they could test whether Jag1 knockdown and/or over-expression rescues Msi
manipulations in scratch wound (see below), or cell scatter assays in*
Figure 6*,
respectively*.

We have directly tested the link between Msi and Jag1 using luciferase reporters. We
showed that translation of a reporter with the Jag1 3' UTR is enhanced by knockdown
of Msi1/Msi2 in 293T cells. These experiments strengthen the link between Msi and Jag1
and also demonstrate that the regulation of Jag1 by Msi is conserved in human cells. The
responses to MPs 7 & 8 below provide more information. In addition, we now show
that Jag1 protein expression is reduced in mouse mammary glands following overexpression
of Msi2.

4) The authors should provide more information on how they define Msi
translation targets. How many genes were considered as targets? What fraction of
genes have 3' UTRs enriched in UAGs (in the 85% percentile range) that are not
regulated by Msi?

We have included more information in the manuscript on how translational targets were
defined using filters to eliminate genes with low read coverage or large mRNA level
changes and requiring a minimum 3-fold change in TE. The majority of genes containing
the UAG motif in the 3' UTR are not translationally regulated by Msi in NPCs, since
only a small number of genes were differentially translated, while the UAG motif is
relatively common in 3' UTR regions. For example, only 39 genes in were
differentially expressed in the Msi1 overexpression experiments, out of which only a
handful of genes showed very large changes in TE. It is possible that co-factors are
required in vivo for Msi to affect translation following binding to the mRNA, or that
other RNA-binding factors outcompete Msi protein for binding. The molecular mechanism
underlying Musashi-dependent translational control and the nature of any co-factors
involved are not known.

*5) The changes in splicing are proposed to be indirect, in large part because
the bulk of Msi proteins are cytoplasmic. However, a possible direct role should be
acknowledged in the absence of additional data*.

We agree that we cannot exclude based on current data that Msi proteins directly affect
splicing and have added this point to the Discussion.

*6) It is an overstatement to say that Msi OE inhibits wound healing. Wound
healing is a complex process that is tightly regulated and involves multiple layers
of tissues acting in a coordinated way. To say that Msis inhibit wound healing may be
taken to suggest that the authors have observed Msis actually acting in the process
of wound healing rather than in the in vitro phenotypic scratch wound assay. The
authors should change their language from “wound healing” to
“migration”*.

We agree with the reviewers that “migration” is more precise than
“wound healing” given our data. We have changed the text to use
“migration” in place of “wound healing”, and used a new
assay to specifically test migration (see response to MPs 7 & 8 below).

*7) The authors should verify induction of EMT upon knock down of Msis in the
cell scatter assays in*
Figure 6*, and recovery
of an epithelial phenotype upon Msi expression in the scratch assays*.

Response is combined with response to main point 8 below.

*8) Msi OE causes a slight delay in cell migration as it may also do to mammary
ductal branching. The defect in mammary ductal branching is actually unclear (is it
amplitude, number of branches?). Where possible, quantification should be applied.
The authors propose that mammary gland development is a « a type of EMT ».
References linking mammary development to EMT should be provided*.

Response to MPs #7 and #8. We verified induction of EMT upon knockdown of Msis
in cancer cell lines, where the EMT transition is most relevant. Our new data (Figure 6) show that epithelial markers are generally
downregulated while mesenchymal markers are upregulated upon knockdown of Msis in an
epithelial cancer cell line. To further solidify this connection, we overexpressed Msi1
in a mesenchymal cell line (MDAMB231) where Msi1 levels were initially extremely low. We
found that Msi1 overexpression led to a decrease in mesenchymal markers and an increase
in epithelial markers, consistent with our model that Msi proteins promote an epithelial
state. To address reviewer comments regarding cell migration, we performed a migration
transwell assay in breast cancer cell lines. We found that overexpression of Msi1 in the
mesenchymal cell line LM2 (a derivative of MDAMB231) strongly impaired migration in the
transwell assay (Figure 6). These migration
assays are more quantitative and controlled than the scratch assay we performed
previously, and we feel that the use of breast cancer cell lines for migration analysis
is particularly relevant, given the roles for EMT in breast cancer.

We added quantitation and additional explanation of the mammary ductal branching
phenotype that occurs upon Msi2 overexpression (Figure 7—figure supplement 1). The results show that Msi2 overexpression
reduces the number of ductal branch points by approximately twofold at both 4 weeks and
7 weeks following induction of Msi2 (both P < 0.01 by t-test), and that Msi2
overexpression also delays ductal branch growth (P < 0.01 at 4 weeks, but
indistinguishable from control at 7 weeks).

*9) Typically, EMT has been linked to metastasis. It would be most relevant to
test the impact of Msi OE on the metastatic potential of cancer cells injected into
mice*.

We agree that it would be worthwhile to investigate the impact of Musashi proteins on
metastasis in vivo, but we feel that this is beyond the scope of our manuscript. While
we have linked Musashi proteins to regulation of the epithelial state and EMT in cancer
(through functional analysis of cancer cell lines and computational analysis of TCGA
data), and in normal development (through in vivo analyses of healthy mammary gland
development), we do not directly address metastasis in this work. We have been careful
not to imply in the text that our work bears directly on metastasis, or that the role of
Musashi proteins in cancer occurs through their effect on metastatic processes. We
believe that our results on the regulation of the epithelial state (and of EMT in a
healthy mammary gland context) have consequences that are significant independent of
possible effects on metastasis, and may not directly bear on metastasis. In some cancers
like glioblastoma, where Msis are highly expressed, metastases are rare. In breast
cancer, highly proliferative epithelial tumors (like Luminal B type tumors) can be
aggressive and harmful to patients without possessing the mesenchymal properties that
lead to metastases. Therefore, we feel that potential links to metastasis are very
worthwhile to explore, but are not essential to the conclusions of this paper.